# Soft Calibration Objectives for Neural Networks

**Archit Karandikar**[*]
Google Research
archk@google.com

**Nicholas Cain**[*]
Google Research
nicholascain@google.com

**Dustin Tran**
Google Research
trandustin@google.com

**Balaji Lakshminarayanan**
Google Research
balajiln@google.com

**Jonathon Shlens**
Google Research
shlens@google.com

**Michael C. Mozer**
Google Research
mcmozer@google.com

**Becca Roelofs**
Google Research
rolfs@google.com

## Abstract

Optimal decision making requires that classifiers produce uncertainty estimates consistent with their empirical accuracy. However, deep neural networks are often under- or over-confident in their predictions. Consequently, methods have been developed to improve the calibration of their predictive uncertainty, both during training and post-hoc. In this work, we propose differentiable losses to improve calibration based on a soft (continuous) version of the binning operation underlying popular calibration-error estimators. When incorporated into training, these soft calibration losses achieve state-of-the-art single-model ECE across multiple datasets with less than 1% decrease in accuracy. For instance, we observe an 82% reduction in ECE (70% relative to the post-hoc rescaled ECE) in exchange for a 0.7% relative decrease in accuracy relative to the cross-entropy baseline on CIFAR-100. When incorporated post-training, the soft-binning-based calibration error objective improves upon temperature scaling, a popular recalibration method. Overall, experiments across losses and datasets demonstrate that using calibration-sensitive procedures yield better uncertainty estimates under dataset shift than the standard practice of using a cross-entropy loss and post-hoc recalibration methods.[2]

## 1 Introduction

Despite the success of deep neural networks across a variety of domains, they are still susceptible to miscalibrated predictions. Both over- and under-confidence contribute to miscalibration, and empirically, deep neural networks empirically exhibit significant miscalibration [Guo et al., 2017]. Calibration error (*CE*) quantifies a model's miscalibration by measuring how much its confidence, i.e. the predicted probability of correctness, diverges from its accuracy, i.e. the empirical probability of correctness. Models with low CE are critical in domains where satisfactory outcomes depend on well-modeled uncertainty, such as autonomous vehicle navigation [Bojarski et al., 2016] and medical diagnostics [Jiang et al., 2012, Caruana et al., 2015, Kocbek et al., 2020]. Calibration has also been shown to be useful for improving model fairness [Pleiss et al., 2017] and detecting out-of-distribution data [Kuleshov and Ermon, 2017, Devries and Taylor, 2018, Shao et al., 2020]. More generally, low

---

[*]co-first author

[2]Code available on GitHub: https://github.com/google/uncertainty-baselines/tree/main/experimental/caltrain

CE is desirable in any setting in which thresholds are applied to the predicted confidence of a neural network in order to make a decision.

Methods for quantifying CE typically involve binning model predictions based on their confidence. CE is then computed empirically as a weighted average of the absolute difference in average prediction confidence and average accuracy across different bins [Naeini et al., 2015]. Oftentimes these bins are selected heuristically such as *equal-width* (uniformly binning the score interval) and *equal-mass* (with equal numbers of samples per-bin) [Nixon et al., 2019].

However, these commonly used measures of CE are not trainable with gradient-based methods because the binning operation is discrete and has zero derivatives. As a result, neural network parameters are not directly trained to minimize CE, either during training or during post-hoc recalibration. In this paper, we introduce new objectives based on a differentiable binning scheme that can be used to efficiently and directly optimize for calibration.

**Contributions.** We propose estimating CE with soft (i.e., overlapping, continuous) bins rather than the conventional hard (i.e., nonoverlapping, all-or-none) bins. With this formulation, the CE estimate is differentiable, allowing us to use it as (1) a secondary (i.e., auxiliary) loss to incentivize model calibration during training, and (2) a primary loss for optimizing post-hoc recalibration methods such as temperature scaling. In the same spirit, we soften the AvUC loss [Krishnan and Tickoo, 2020], allowing us to use it as an effective secondary loss during training for non-Bayesian neural networks where the AvUC loss originally proposed for Stochastic Variational Inference (SVI) typically does not work. Even when training with the cross-entropy loss results in training set memorization (perfect train accuracy and calibration), Soft Calibration Objectives are still useful as secondary training losses for reducing test ECE using a procedure we call *interleaved training*.

In an extensive empirical evaluation, we compare Soft Calibration Objectives as secondary losses to existing calibration-incentivizing losses. In the process, we find that soft-calibration losses outperform prior work on in-distribution test sets. Under distribution shift, we find that calibration-sensitive training objectives as a whole (not always the ones we propose) result in better uncertainty estimates compared to the standard cross-entropy loss coupled with temperature scaling.

Our contributions can be summarized as follows:

- We propose simple Soft Calibration Objectives S-AvUC, SB-ECE as secondary losses which optimize for CE *throughout training*. We show that across datasets and choice of primary losses, the S-AvUC secondary loss results in the largest improvement in ECE as per the Cohen's $d$ effect-size metric (Figure 1). We also show that such composite losses obtain *state-of-the-art single-model ECE* in exchange for less than 1% reduction in accuracy (Figure 5) for CIFAR-10, CIFAR-100, and Imagenet.

- We improve upon temperature scaling - a popular post-hoc recalibration method - by directly optimizing the temperature parameter for soft calibration error instead of the typical log-likelihood. Our extension (TS-SB-ECE) consistently beats original temperature scaling (TS) across different datasets, loss functions and calibration error measures, and we find that the performance is better under dataset shift (Figure 2).

- Overall, our work demonstrates a fundamental advantage of objectives which better incentivize calibration over the standard practice of training with cross-entropy loss and then applying post-hoc methods such as temperature scaling. Uncertainty estimates of neural networks trained using these methods generalize better in and out-of-distribution.

## 2 Related Work

Many techniques have been proposed to train neural networks for calibration. These can be organized into three categories. One category augments or replaces the primary training loss with a term to explicitly incentivize calibration. These include the AvUC loss [Krishnan and Tickoo, 2020], MMCE loss [Kumar et al., 2018] and Focal loss [Mukhoti et al., 2020, Lin et al., 2018]. The Mean Squared Error loss also compares favourably [Hui and Belkin, 2021] to cross-entropy loss. We show that our methods outperform all these calibration-incentivizing training objectives, applied across multiple primary losses. Label smoothing [Müller et al., 2020] has been shown to improve calibration and can be interpreted as a modified primary loss.

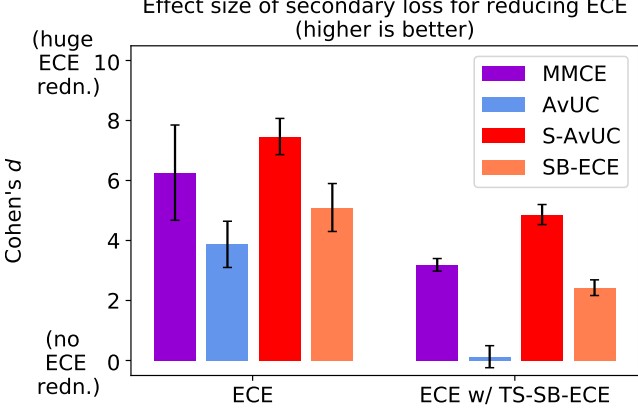

Figure 1: We compare the effect size of various secondary losses on ECE (equal-mass binning, $\ell_2$ norm) across datasets and primary losses, both with and without post-hoc temperature scaling. The S-AvUC secondary loss we propose shows the strongest positive effect, followed by MMCE and SB-ECE. Note that a d-value of 0.8 (resp., 2.0) is considered a large (resp., huge) positive effect and d-values obtained here are much larger. Secondary losses which incentivize calibration show strong positive effect for reducing ECE even after temperature scaling.

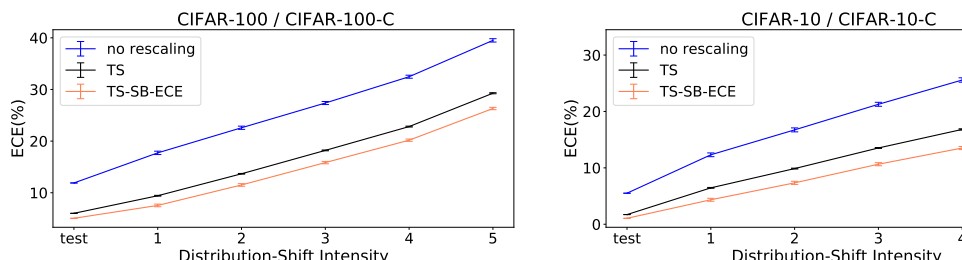

Figure 2: Post-hoc temperature scaling with the soft calibration error objective (TS-SB-ECE) reduces ECE more than standard post-hoc temperature scaling (TS), particularly under distribution shift. This result holds across datasets (left and right panels), distribution shift intensities (along abscissa) and training objectives (not shown). The training objective shown here for both datasets is the most popular one: NLL. The ECE value (equal-mass binning, $\ell_2$ norm) shown is the mean ECE across the corruption types that constitute CIFAR-10-C and CIFAR-100-C. Error bars are $\pm 1$ standard error of mean (SEM), corrected for intrinsic variability due to type of corruption [Masson and Loftus, 2003].

A second category of methods are post-hoc calibration methods, which rescale model predictions after training. These methods optimize additional parameters on a held-out validation set [Platt, 1999, Zadrozny and Elkan, 2002, Kull et al., 2019, Zadrozny and Elkan, 2001, Naeini and Cooper, 2016, Allikivi and Kull, 2019, Kull et al., 2017, Naeini et al., 2015, Wenger et al., 2020, Gupta et al., 2020]. The most popular technique is temperature scaling [Guo et al., 2017], which maximizes a single temperature parameter on held-out NLL. We examine temperature scaling and propose an improvisation that directly optimizes temperature for a soft calibration objective instead of NLL. Temperature scaling has shown to be ineffective under distribution shift in certain scenarios [Ovadia et al., 2019]. We show that uncertainty estimates of methods which train for calibration generalize better than temperature scaling under distribution shift.

A third category of methods examines model changes such as ensembling multiple predictions [Lakshminarayanan et al., 2017, Wen et al., 2020] or priors [Dusenberry et al., 2020]. Similar to previous work [Kumar et al., 2018, Lin et al., 2018], we focus on the choice of loss functions for improving calibration of a single neural network—whether during training or post-hoc—and do not compare against ensemble models or Bayesian neural networks. These techniques are complementary to ours and can be combined with our techniques to further improve performance.

Recent works have investigated issues [Nixon et al., 2019, Kumar et al., 2019, Roelofs et al., 2020, Gupta et al., 2020] with the originally proposed ECE [Guo et al., 2017] and suggested new ones. Debiased CE [Kumar et al., 2019] and mean-sweep CE [Roelofs et al., 2020] have been shown to have lesser bias and more consistency across the number of bins parameter than ECE whereas KS-error [Gupta et al., 2020] avoids binning altogether. We report these metrics in the Appendix.

# 3 Background

## 3.1 The Task and the Model

Consider a classification task over $K$ classes with a dataset of $N$ samples $D = \langle(\boldsymbol{x}_i, y_i)\rangle_{i=1}^N$ drawn from the joint probability distribution $\mathcal{D}(\mathcal{X}, \mathcal{Y})$ over the input space $\mathcal{X}$ and label space $\mathcal{Y} = \{1, 2, \ldots K\}$. The task is modelled using a deep neural network with parameters $\boldsymbol{\theta}$ whose top layer is interpreted as a softmax layer. The top layer consists of $K$ neurons which produce logits $\boldsymbol{g_\theta}(\boldsymbol{x}) = \langle g_\theta(y|\boldsymbol{x})\rangle_{y\in\mathcal{Y}}$. The predictive probabilities for a given input are:

$$\boldsymbol{f_\theta}(\boldsymbol{x}) = \langle f_\theta(y|\boldsymbol{x})\rangle_{y\in\mathcal{Y}} = \text{softmax}(\boldsymbol{g_\theta}(\boldsymbol{x})).$$

The parameters $\boldsymbol{\theta}$ of the neural network are trained to minimize $\mathbb{E}_{(\boldsymbol{x},y)}\mathcal{L}(\boldsymbol{f_\theta}(\boldsymbol{x}), y)$ where $(\boldsymbol{x}, y)$ is sampled from $\mathcal{D}(\mathcal{X}, \mathcal{Y})$. Here $\mathcal{L}$ is a trainable loss function which incentivizes the predictive distribution $\boldsymbol{f_\theta}(\boldsymbol{x})$ to fit the label $y$. The model's prediction on datapoint $(\boldsymbol{x}, y)$ is denoted by $q_\theta(\boldsymbol{x}) = \arg\max(\boldsymbol{f_\theta}(\boldsymbol{x}))$. We denote by $c_\theta(\boldsymbol{x}) = \max(\boldsymbol{f_\theta}(\boldsymbol{x}))$ the confidence of this prediction and by boolean quantity $a_\theta(\boldsymbol{x}, y) = \mathbf{1}_{q_\theta(\boldsymbol{x})=y}$ the accuracy of this prediction.

## 3.2 Expected Calibration Error (ECE)

Given a distribution $\hat{\mathcal{D}}(\mathcal{X}, \mathcal{Y})$ on datapoints, there are two standard notions of Ideal Calibration Error that we refer to as Ideal Binned Expected Calibration Error (of order $p$), $\text{IECE}_{\text{bin},p}(\hat{\mathcal{D}}, \boldsymbol{\theta}) = \text{IECE}_{\text{bin},p}$, and Ideal Expected Label-Binned Calibration Error (of order $p$), $\text{IECE}_{\text{lb},p}(\hat{\mathcal{D}}, \boldsymbol{\theta}) = \text{IECE}_{\text{lb},p}$. This nomenclature is consistent with that introduced by Roelofs et al. [2020]. Both these measure, in slightly different ways, the $p^{\text{th}}$ root of the $p^{\text{th}}$ moment of the absolute difference between model confidence and the empirical accuracy given that confidence. They are defined as follows:

$$\text{IECE}_{\text{bin},p}(\hat{\mathcal{D}}, \boldsymbol{\theta}) = \left(\mathbb{E}_{c_\theta(\hat{\boldsymbol{x}}_0)}\left[|\mathbb{E}[a_\theta(\hat{\boldsymbol{x}}_1, \hat{y}_1)|c_\theta(\hat{\boldsymbol{x}}_1) = c_\theta(\hat{\boldsymbol{x}}_0)] - c_\theta(\hat{\boldsymbol{x}}_0)|^p\right]\right)^{1/p} \tag{1}$$

$$\text{IECE}_{\text{lb},p}(\hat{\mathcal{D}}, \boldsymbol{\theta}) = \left(\mathbb{E}_{(\hat{\boldsymbol{x}}_0, \hat{y}_0)}\left[|\mathbb{E}[a_\theta(\hat{\boldsymbol{x}}_1, \hat{y}_1)|c_\theta(\hat{\boldsymbol{x}}_1) = c_\theta(\hat{\boldsymbol{x}}_0)] - c_\theta(\hat{\boldsymbol{x}}_0)|^p\right]\right)^{1/p}. \tag{2}$$

Note that the critical dependence on $\hat{\mathcal{D}}(\mathcal{X}, \mathcal{Y})$ is implicit in both definitions since the datapoint $(\hat{\boldsymbol{x}}_0, \hat{y}_0)$ from the outer expectation and the datapoint $(\hat{\boldsymbol{x}}_1, \hat{y}_1)$ from the inner expectation are both sampled from $\hat{\mathcal{D}}(\mathcal{X}, \mathcal{Y})$.

We cannot compute $\text{IECE}_{\text{bin},p}$ and $\text{IECE}_{\text{lb},p}$ in practice since the number of datapoints are finite. Instead we consider a dataset $\hat{D} = \langle(\hat{\boldsymbol{x}}_i, \hat{y}_i)\rangle_{i=1}^{\hat{N}}$ drawn from $\hat{\mathcal{D}}(\mathcal{X}, \mathcal{Y})$ and partition the confidence interval $[0, 1]$ into bins $\mathcal{B} = \langle B_i\rangle_{i\in\{1,2,\ldots M\}}$, each of which also corresponds to a confidence interval. We will use $c_i$ as a shorthand for $c_\theta(\hat{\boldsymbol{x}}_i)$ and $a_i$ as a shorthand for $a_\theta(\hat{\boldsymbol{x}}_i, \hat{y}_i)$. We denote by $b_i(\mathcal{B}, \hat{D}, \boldsymbol{\theta}) = b_i$ the bin to which $c_i$ belongs. We define the size of bin $j$ as $S_j(\mathcal{B}, \hat{D}, \boldsymbol{\theta}) = S_j$, the average confidence of bin $j$ as $C_j(\mathcal{B}, \hat{D}, \boldsymbol{\theta}) = C_j$ and the average accuracy of bin $j$ as $A_j(\mathcal{B}, \hat{D}, \boldsymbol{\theta}) = A_j$. These are expressed as follows:

$$S_j(\mathcal{B}, \hat{D}, \boldsymbol{\theta}) = |\{i|b_i = j\}| \tag{3}$$

$$C_j(\mathcal{B}, \hat{D}, \boldsymbol{\theta}) = \tfrac{1}{S_j}\Sigma_{i|b_i=j}c_i \tag{4}$$

$$A_j(\mathcal{B}, \hat{D}, \boldsymbol{\theta}) = \tfrac{1}{S_j}\Sigma_{i|b_i=j}a_i. \tag{5}$$

We are now in a position to define the Expected Binned Calibration Error of order $p$ which we denote by $\text{ECE}_{bin,p}$ and Expected Label-Binned Calibration Error of order $p$ we denote by $\text{ECE}_{lb,p}$. These serve as empirical approximations to the corresponding intractable ideal notions from equations 1 and 2. They are defined as follows:

$$\text{ECE}_{bin,p}(\mathcal{B}, \hat{D}, \boldsymbol{\theta}) = \left(\Sigma_{i=1}^M \tfrac{S_j}{\hat{N}} \cdot |A_j - C_j|^p\right)^{1/p} \tag{6}$$

$$\text{ECE}_{lb,p}(\mathcal{B}, \hat{D}, \boldsymbol{\theta}) = \left(\tfrac{1}{\hat{N}}\Sigma_{i=1}^{\hat{N}}|A_{b_i} - c_i|^p\right)^{1/p}. \tag{7}$$

It follows from Jensen's inequality that $\text{ECE}_{lb,p}(\mathcal{B}, D, \boldsymbol{\theta}) \geq \text{ECE}_{bin,p}(\mathcal{B}, D, \boldsymbol{\theta})$ [Roelofs et al., 2020].

# 4 Soft Calibration Objectives

In this section, we define quantities that can be used to better incentivize calibration during training.

## 4.1 Soft-Binned ECE (SB-ECE)

The quantities in the definitions of $\text{ECE}_{bin,p}$ and $\text{ECE}_{lb,p}$ can be written in terms of a formal definition of the bin membership function. Let us denote the bin-membership function for a given binning $\mathcal{B} = \langle B_i \rangle_{i \in \{1,2,\dots M\}}$ by $\boldsymbol{u}_{\mathcal{B}} : [0,1] \to \mathcal{U}_M$, where $\mathcal{U}_M = \{\boldsymbol{v} \in [0,1]^M | \Sigma_j v_j = 1\}$ is the set of possible bin membership vectors over $M$ bins. The membership function for bin $j$ is denoted by $u_{\mathcal{B},j} : [0,1] \to [0,1]$ and is defined by $u_{\mathcal{B},j}(c) = \boldsymbol{u}_{\mathcal{B}}(c)_j$. The size, average accuracy, and average confidence of bin $j$ from equations 3, 4, and 5 can now be written in terms of $\boldsymbol{u}_{\mathcal{B}}$ as follows:

$$S_j(\mathcal{B}, \hat{D}, \boldsymbol{\theta}) = \Sigma_{i=1}^{\hat{N}} u_{\mathcal{B},j}(c_i) \tag{8}$$

$$C_j(\mathcal{B}, \hat{D}, \boldsymbol{\theta}) = \frac{1}{S_j} \Sigma_{i=1}^{\hat{N}} (u_{\mathcal{B},j}(c_i) \cdot c_i) \tag{9}$$

$$A_j(\mathcal{B}, \hat{D}, \boldsymbol{\theta}) = \frac{1}{S_j} \Sigma_{i=1}^{\hat{N}} (u_{\mathcal{B},j}(c_i) \cdot a_i). \tag{10}$$

The quantities $\text{ECE}_{bin,p}$ and $\text{ECE}_{lb,p}$ can further be written in terms of the quantities $S_j$, $C_j$ and $A_j$ using equation 6 and a modification of equation 7 (see equation 12 below). We know that the differentials $\partial \text{ECE}_{bin,p} / \partial \boldsymbol{\theta}$ and $\partial \text{ECE}_{lb,p} / \partial \boldsymbol{\theta}$ are non-trainable. The formulation above makes it clear that this is precisely because $\partial u_{\mathcal{B},j} / \partial c$ is zero within bin boundaries and undefined at bin boundaries. Moreover, this observation implies that if we could come up with a trainable soft bin-membership function $\boldsymbol{u}_{\mathcal{B}}^*$ then we could use it in place of the usual hard bin-membership function $\boldsymbol{u}_{\mathcal{B}}$ to obtain a trainable version of $\text{ECE}_{bin,p}$ and $\text{ECE}_{lb,p}$.

With this motivation, we define the soft bin-membership function that has a well-defined non-zero gradient in $(0,1)$. It is parameterized by the number of bins $M$ and a temperature parameter $T$. We consider equal-width binning for simplicity and so we represent it as $\boldsymbol{u}_{M,T}^*$ rather than $\boldsymbol{u}_{\mathcal{B}}^*$. We desire unimodality over confidence: if $\xi_j$ denotes the center of bin $j$ then we want $\partial u_{M,T,j}^* / \partial c$ to be positive for $c < \xi_j$ negative for $c > \xi_j$. Similarly, we also desire unimodality over bins: if $c < \xi_i < \xi_j$ or $c > \xi_i > \xi_j$, then we want that $u_{M,T,i}^*(c) > u_{M,T,j}^*(c)$. Finally, we also want the aforementioned temperature parameter $T$ to control how close the binning is to hard binning (i.e. how steeply membership drops off). This would give us the nice property of hard-binning being a limiting condition of soft-binning. With this motivation, we define the soft bin-membership function as

$$\boldsymbol{u}_{M,T}^*(c) = \text{softmax}(\boldsymbol{g}_{M,T}(c)),$$
$$\text{where } g_{M,T,i}(c) = -(c - \xi_i)^2 / T \qquad \forall i \in \{1, 2, \dots M\}.$$

Figure 3 visualizes soft bin-membership. We can now formulate trainable calibration error measures. We define the Expected Soft-Binned Calibration Error $\text{SB-ECE}_{\text{bin},p}(M, T, \hat{D}, \boldsymbol{\theta})$ and Expected Soft-Label-Binned Calibration Error $\text{SB-ECE}_{\text{lb},p}(M, T, \hat{D}, \boldsymbol{\theta})$:

$$\text{SB-ECE}_{\text{bin},p}(M, T, \hat{D}, \boldsymbol{\theta}) = \left( \Sigma_{i=1}^M (\frac{S_j}{\hat{N}} |A_j - C_j|^p) \right)^{1/p}, \tag{11}$$

$$\text{SB-ECE}_{\text{lb},p}(M, T, \hat{D}, \boldsymbol{\theta}) = \left( \frac{1}{\hat{N}} \Sigma_{i=1}^{\hat{N}} \Sigma_{j=1}^M (u_{\mathcal{M},T,j}^*(c_i) \cdot |A_j - c_i|^p) \right)^{1/p}. \tag{12}$$

The quantities $S_j$, $C_j$ and $A_j$ in these expressions are obtained by using the soft bin membership function $\boldsymbol{u}_{M,T}^*$ in place of the hard bin membership function $\boldsymbol{u}_{\mathcal{B}}$ in equations 8, 9 and 10 respectively.

We use these trainable calibration error measures as: (1) part of the training loss function and (2) the objective that is minimized to tune the temperature scaling parameter $T_{ts}$ during post-hoc calibration.

## 4.2 Soft AvUC (S-AvUC)

The accuracy versus uncertainty calibration (AvUC) loss [Krishnan and Tickoo, 2020] categorizes each prediction that a model with parameters $\boldsymbol{\theta}$ makes for a labelled datapoint $d_i = (\boldsymbol{x}_i, y_i)$ from dataset $\hat{D}$ according to two axes: (1) accurate [A] versus inaccurate [I], based on the value of the boolean quantity $a_{\boldsymbol{\theta}}(\boldsymbol{x}_i, y_i) = a_i$ (2) certain [C] versus uncertain [U], based on whether the entropy

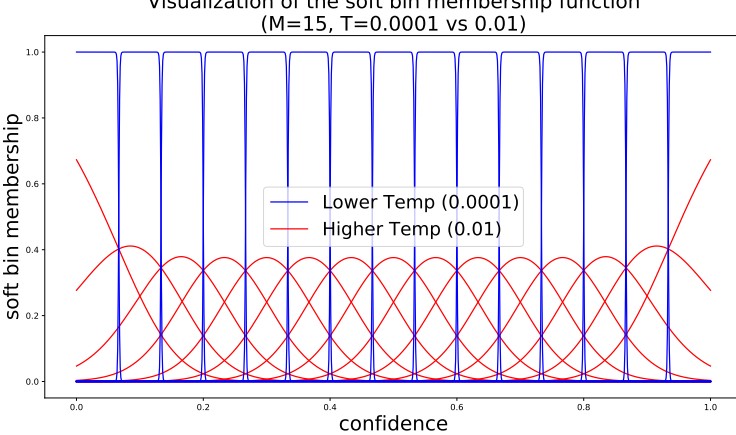

Figure 3: Visualization of the soft bin membership function which shows that the temperature parameter determines the sharpness of the binning. Soft binning limits to hard binning as temperature tends to zero.

$h(f_{\boldsymbol{\theta}}(\boldsymbol{x}_i)) = h_i$ of the predictive distribution is above or below a threshold $\kappa$. Denote the number of elements from $\hat{D}$ that fall in each of these 4 categories by $\hat{n}_{AC}$, $\hat{n}_{AU}$, $\hat{n}_{IC}$ and $\hat{n}_{IU}$ respectively. The AvUC loss incentivizes the model to be certain when accurate and uncertain when inaccurate:

$$\text{AvUC}(\kappa, \hat{D}, \boldsymbol{\theta}) = \log\left(1 + \frac{n_{AU} + n_{IC}}{n_{AC} + n_{IU}}\right), \tag{13}$$

where the discrete quantities are relaxed to be differentiable:

$$n_{AU} = \Sigma_{i|(\boldsymbol{x}_i,y_i)\in S_{AU}}(c_i \tanh h_i) \qquad n_{IC} = \Sigma_{i|(\boldsymbol{x}_i,y_i)\in S_{IC}}((1-c_i)(1-\tanh h_i))$$
$$n_{AC} = \Sigma_{i|(\boldsymbol{x}_i,y_i)\in S_{AC}}(c_i(1-\tanh h_i)) \qquad n_{IU} = \Sigma_{i|(\boldsymbol{x}_i,y_i)\in S_{IU}}((1-c_i)\tanh h_i). \tag{14}$$

Krishnan and Tickoo [2020] have showed good calibration results using the AvUC loss in SVI settings. However, in our experiments we found that the addition of the AvUC loss term resulted in poorly calibrated models in non-SVI neural network settings (see Appendix A). One reason for this seems to be that minimizing the AvUC loss results in the model being incentivized to be even more confident in its inaccurate and certain predictions (via minimizing $n_{IC}$) and even less confident in its accurate and uncertain predictions (via minimizing $n_{AU}$). This conjecture is validated by experimental observations: when we stopped the gradients flowing through the $c_i$ terms in equation 14, we were able to obtain calibrated models (see Appendix F). Fixing this incentivization issue in a more principled manner than stopping gradients is desirable. Another desirable improvisation is replacing the hard categorization into the certain/uncertain bins with a soft partitioning scheme. We meet both these objectives by defining a notion of soft uncertainty.

We want a limiting case of the soft uncertainty function to be the hard uncertainty function based on an entropy threshold $\kappa$. This implies that we will continue to have a parameter $\kappa$ despite getting rid of the hard threshold. As before, we desire a temperature parameter $T$ that will determine how close the function is to the hard uncertainty function. The soft uncertainty function $t_{\kappa,T} : [0,1] \rightarrow [0,1]$ takes as input the $[0,1]$-normalized entropy $h_i^* = h_i / \log(K)$ of the predicted posterior where $K$ is the number of classes. We also need $\partial t_{\kappa,T}/\partial h^*$ to be positive in $[0,1]$ and would like $t_{\kappa,T}$ to satisfy $\lim_{h^* \rightarrow 0} t_{\kappa,T}(h^*) = 0$ and $\lim_{h^* \rightarrow 1} t_{\kappa,T}(h^*) = 1$. Finally, it would be good to have the $[0,1]$ identity mapping as a special case of $t_{\kappa,T}$-family for some value of $\kappa$ and $T$. We now define the soft-uncertainty function in the following way so that it meets all stated desiderata:

$$t_{\kappa,T}(h^*) = \text{logistic}\left(\frac{1}{T}\log\frac{h^*(1-\kappa)}{(1-h^*)\kappa}\right).$$

Finally, we define Soft AvUC in terms of soft uncertainty by modifying equations 13 and 14. In our experiments, we use Soft AvUC as part of the loss function to obtain calibrated models.

$$\text{S-AvUC}(\kappa, T, \hat{D}, \boldsymbol{\theta}) = \log\left(1 + \frac{n'_{AU} + n'_{IC}}{n'_{AC} + n'_{IU}}\right), \tag{15}$$

where

$$n'_{AU} = \Sigma_{i|a_i=1}(t_{\kappa,T}(h_i^*)\tanh h_i) \qquad n'_{IC} = \Sigma_{i|a_i=0}((1-t_{\kappa,T}(h_i^*))(1-\tanh h_i))$$
$$n'_{AC} = \Sigma_{i|a_i=1}((1-t_{\kappa,T}(h_i^*))(1-\tanh h_i)) \qquad n'_{IU} = \Sigma_{i|a_i=0}(t_{\kappa,T}(h_i^*)\tanh h_i). \tag{16}$$

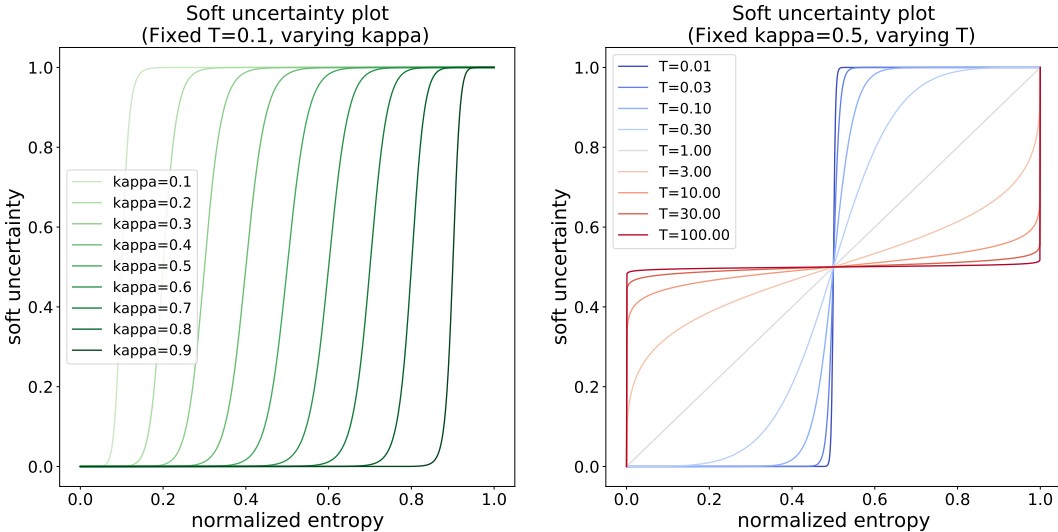

Figure 4: Visualization of the soft uncertainty function $t_{\kappa,T}(h^*)$ which shows that the parameter $\kappa$ captures the soft-threshold whereas the parameter $T$ captures the sharpness of the thresholding.

## 5 Results

We compare our Soft Calibration Objectives to recently proposed calibration-incentivizing training objectives MMCE, focal loss, and AvUC on the CIFAR-10, CIFAR-100, and ImageNet datasets. We evaluate the full cross-product of primary and secondary losses: the options for primary loss are cross-entropy (NLL), focal or mean squared error (MSE) loss; and the options for secondary loss are MMCE, AvUC, SB-ECE or S-AvUC. Results for the MSE primary loss and the AvUC secondary loss are in Appendix A. Our experiments build on the Uncertainty Baselines and Robustness Metrics libraries [Nado et al., 2021, Djolonga et al., 2020].

### 5.1 Soft Calibration Objectives for End-to-End Training

Our results demonstrate that training losses which include Soft Calibration Objectives obtain state-of-the-art single-model ECE on the test set in exchange for less than 1% reduction in accuracy for all three datasets that we experiment with. In fact, our methods (especially S-AvUC) without post-hoc temperature scaling are better than or as good as other methods with or without post-hoc temperature scaling on all three datasets.

The primary losses we work with for CIFAR-10/100 are the cross-entropy (NLL) loss, the focal loss and mean squared error (MSE) loss. Focal loss [Mukhoti et al., 2020] and MSE loss [Hui and Belkin, 2021] have recently shown to outperform the NLL loss in certain settings. The cross-entropy loss outperforms the other two losses on Imagenet, and is thus our sole focus for this dataset.

The primary loss even by itself (especially NLL) can overfit to the train ECE [Mukhoti et al., 2020], without help from the soft calibration losses. Even in such settings, we show that soft calibration losses yield reduction of test ECE using a technique we call 'interleaved training' (Appendix B).

We use the Wide-Resnet-28-10 architecture [Zagoruyko and Komodakis, 2017] trained for 200 epochs on CIFAR-100 and CIFAR-10. For Imagenet, we use the Resnet-50 [He et al., 2015] architecture training for 90 epochs. All our experiments use the SGD with momentum optimizer with momentum fixed to 0.9 and learning rate fixed to 0.1. The loss function we use in our experiments is $\text{PL} + \beta \cdot \text{SL} + \lambda \cdot \text{L2}$ where PL and SL denote the primary and secondary losses respectively and L2 denotes the weight normalization term with $\ell_2$ norm. We tune the $\beta$ and $\lambda$ parameters along with the parameters $\kappa$ and $T$ relevant to the secondary losses SB-ECE$_{lb,p}(M, T, \hat{D}, \boldsymbol{\theta})$ and S-AvUC$(\kappa, T, \hat{D}, \boldsymbol{\theta})$. We tune these hyperparameters sequentially. We fix the learning rate schedule and the number of bins $M$ to keep the search space manageable. Appendix G has more details of our hyperparameter search.

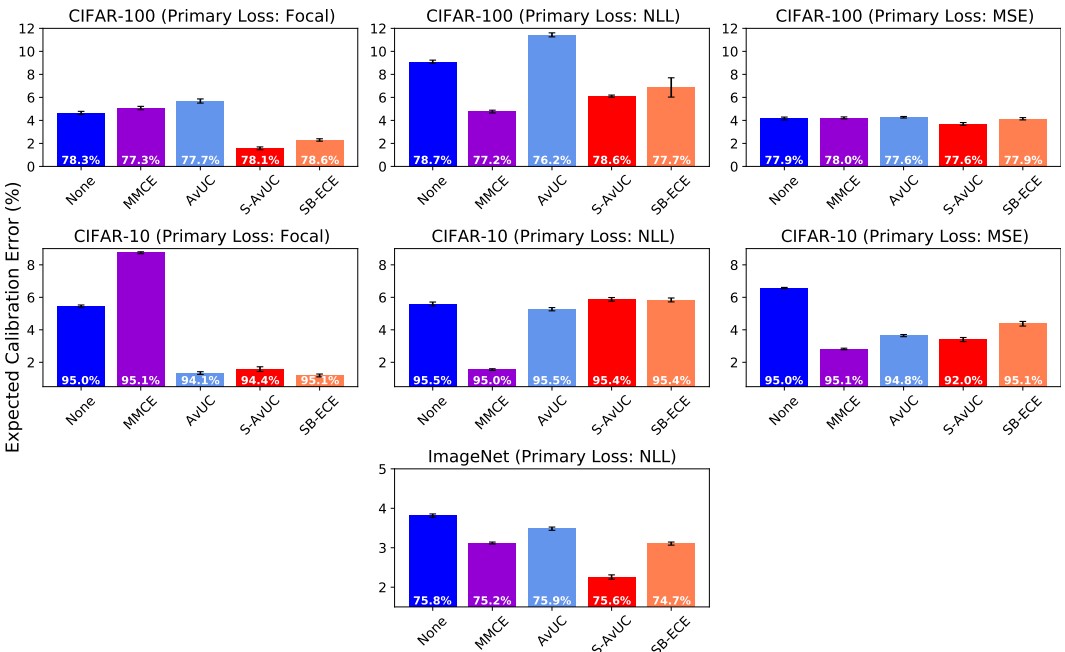

Figure 5: Soft Calibration Objectives (S-AvUC, SB-ECE), when used as secondary losses with a primary loss, achieve lower ECE (equal-mass binning, $\ell_2$ norm) than the corresponding primary loss for CIFAR-10, CIFAR-100, and ImageNet. These statistically significant wins come at the cost of less than 1% accuracy (reported at bottom of the bar). Values reported are mean over 10 runs, and the error bars indicate $\pm 1$ standard error of the mean (SEM). For each dataset (each row) the best (across primary losses) ECE obtained using the S-AvUC and SB-ECE secondary losses is lower than the best ECE obtained using existing techniques.

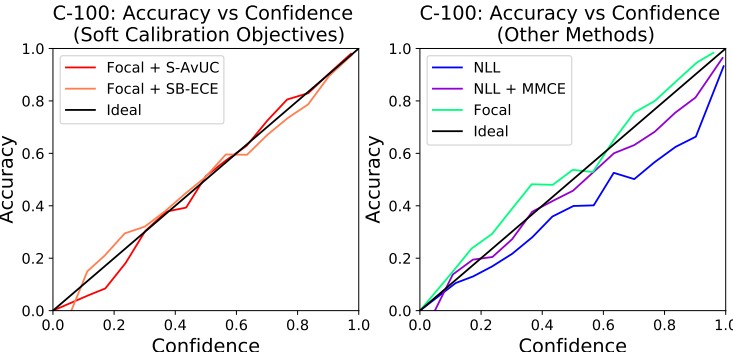

Figure 6: Accuracy vs Confidence plots for various methods on CIFAR-100. NLL is significantly overconfident and NLL + MMCE is somewhat overconfident. While Focal loss is underconfident, augmenting it with Soft Calibration Objectives fixes this issue, resulting in curves closest to the ideal.

In Table 1, we report the runs with the best ECE values that also came within 1% of the primary loss run with the highest accuracy. Figure 6 is the accuracy-confidence plot corresponding to Table 1a. In Figure 5, we visualize the ECE on the test set for all combinations of primary loss, secondary loss and dataset. The best ECE for each dataset is attained using Soft Calibration Objectives as secondary losses. More such figures are in Appendix D and the complete table can be found in Appendix A.

## 5.2 Soft Calibration Objectives for Post-Hoc Calibration

Standard temperature scaling (TS) uses a cross-entropy objective to optimize the temperature. However, using our differentiable soft binning calibration objective (SB-ECE), we can optimize the temperature using a loss function designed to directly minimize calibration error. In Figure 2 (and Figure 9 in Appendix C), we compare temperature scaling with a soft binning calibration objective (TS-SB-ECE) to standard temperature scaling with a cross-entropy objective (TS) on out-of-distribution shifts of increasing magnitude on both CIFAR-10 and CIFAR-100. The distribution

Table 1: We report average accuracy (with standard error across 10 trials), ECE, and ECE obtained after post-hoc temperature scaling (TS) for models trained with different objectives on the CIFAR-10, CIFAR-100, and ImageNet datasets. ECE is computed with the $\ell_2$ norm and equal-mass binning. We find Soft Calibration Objectives (SB-ECE, S-AvUC) result in better or equivalent ECEs compared to previous methods with or without TS. We also find that TS does not always improve ECE. The best ECE value is highlighted for each dataset. The best ECE with TS value is highlighted if it improves over the best value from the ECE column.

(a) CIFAR-100

| Loss Fn. | Accuracy | ECE | ECE with TS |
|---|---|---|---|
| NLL | 78.7±0.122 | 9.10±0.139 | 5.36±0.091 |
| NLL + MMCE | 77.2±0.072 | 4.77±0.121 | 4.06±0.138 |
| Focal | 78.3±0.086 | 4.66±0.130 | 6.47±0.140 |
| Focal + SB-ECE | 78.6±0.062 | 2.30±0.105 | 5.16±0.108 |
| Focal + S-AvUC | 78.1±0.084 | **1.57**±0.122 | 4.15±0.090 |

(b) CIFAR-10

| Loss Fn. | Accuracy | ECE | ECE with TS |
|---|---|---|---|
| NLL | 95.5±0.040 | 5.59±0.119 | 1.95±0.127 |
| NLL + MMCE | 95.0±0.031 | 1.55±0.053 | **1.09**± 0.098 |
| Focal | 95.0± 0.085 | 5.45±0.079 | 2.69±0.190 |
| Focal + SB-ECE | 95.1± 0.056 | **1.19**± 0.088 | 2.08± 0.143 |
| Focal + S-AvUC | 94.4± 0.145 | 1.58± 0.146 | 1.34± 0.172 |

(c) ImageNet

| Loss Fn. | Accuracy | ECE | ECE with TS |
|---|---|---|---|
| NLL | 75.8± 0.036 | 3.81± 0.043 | 2.17± 0.045 |
| NLL + MMCE | 75.2± 0.048 | 3.12± 0.025 | 2.18± 0.029 |
| NLL + SB-ECE | 74.7 ± 0.028 | 3.11 ± 0.039 | **1.92**± 0.024 |
| NLL + S-AvUC | 75.6 ± 0.053 | **2.26** ± 0.055 | 2.02 ± 0.041 |

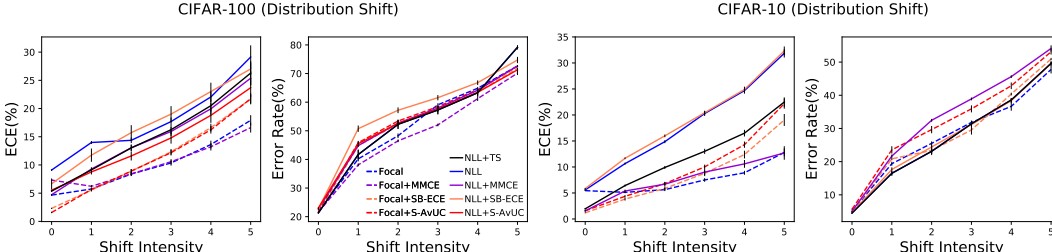

Figure 7: For the CIFAR-10/100 datasets, methods which train for calibration outperform the popular methods - NLL and NLL + TS - under distribution shift. Focal primary loss and MMCE secondary loss result in the lowest ECEs under shift. We note that these methods start off with worse ECEs than our SB-ECE and S-AvUC methods on the test set but end up with better ECE under increasing levels of skew. The OOD datasets that we have used here are CIFAR-10/100-C with skew levels 1-5.

shifts come from either the CIFAR-10-C or CIFAR-100-C datasets. Whereas Figure 2 focuses on cross-entropy loss, Figure 9 also has the plots for other primary losses. Table 2 contains comparisons between the two methods based on test ECE for all combinations of dataset, primary loss and secondary loss. We find that TS-SB-ECE outperforms TS under shift in most cases, and the performance increase is similar across shifts of varying magnitude. Note that temperature scaling (TS) does not always improve ECE, especially when the training loss is different from NLL. In such cases TS-SB-ECE still outperforms TS but may or may not result in ECE improvement.

## 5.3 Training for Calibration Under Distribution Shift

In previous sections we have shown that training for calibration outperforms the popular cross-entropy loss coupled with post-hoc TS on the in-distribution test set. We find that methods which train for calibration (not always our proposed methods) also outperform the cross-entropy loss with TS under dataset shift. Prior work has shown that temperature scaling can perform poorly under distribution shift [Ovadia et al., 2019], and our experiments reproduce this issue. Moreover, we show that training for calibration makes progress towards fixing this problem. However, different methods perform best under distribution shift on different datasets. Whereas S-AvUC does well on ImageNet OOD (see figure 8), Focal loss does better than our methods on CIFAR-10-C and CIFAR-100-C (see figure 7). We cannot prescribe one method in particular under distribution shift given these results but we have shown a crucial benefit of using methods to train for calibration as a whole.

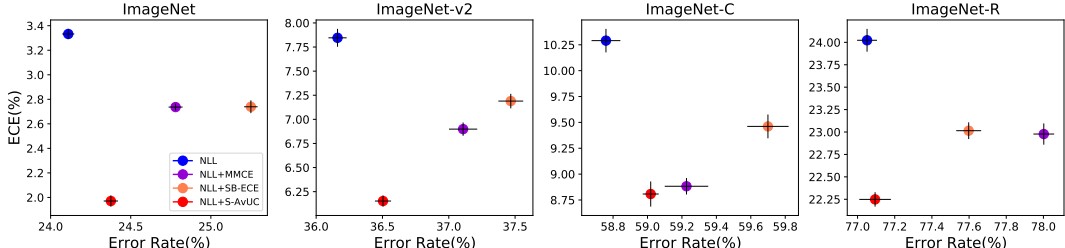

Figure 8: A comparison of methods for ImageNet (primary loss: cross-entropy) shows that the S-AvUC secondary loss yields lowest ECEs under dataset shift. Error bars are ±1 SEM over 10 runs.

## 6  Conclusions

We proposed Soft Calibration Objectives motivated by the goal of directly training models for calibration. These objectives - SB-ECE and S-AvUC - are softened versions of the usual ECE measure and the recently-proposed AvUC loss, respectively. They are easy-to-implement augmentations to the popular cross-entropy loss. We performed a thorough comparison of existing methods of training for calibration. Our experiments show that methods based on soft-calibration objectives can be used to obtain the best ECE among such methods in exchange for less than 1% drop in accuracy. We note that a model being better calibrated overall does not necessarily mean that it is better calibrated for every group and hence the fairness of our methods as well as related methods must be studied. However, our methods of training-for-calibration can be adapted to encourage fairness by applying the methods separately to each protected group.

Even when one does not wish to incorporate secondary losses to train for calibration, we showed that post-hoc temperature scaling works better when tuned using the SB-ECE objective instead of the standard cross-entropy loss. Practitioners can easily replace the cross-entropy loss with our SB-ECE loss when performing post-hoc temperature scaling.

Finally, we demonstrated that the uncertainty estimates of methods which train for calibration generalize better under dataset shift as compared to post-hoc calibration, which is a fundamental motivation for transitioning to training for calibration.

## Acknowledgements

The authors thank Brennan McConnell and Mohammad Khajah who conducted initial explorations of soft binning calibration loss. The authors also thank Zack Nado, D. Sculley and Jeremiah Liu for help with implementation and suggestions for the writeup.

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
