## A   Complete Table of Results

We perform thorough experimentation for the full cross-product of primary and secondary losses for each dataset. We tune several hyperparameters (section G) for each of these settings. For the CIFAR-100 and CIFAR-10 datasets we consider three primary loss functions: cross-entropy, MSE, and Focal. For ImageNet, we consider only the cross-entropy primary loss since the other two did not match its performance in our experiments. We consider five secondary losses: MMCE [Kumar et al., 2018], AvUC [Krishnan and Tickoo, 2020], AvUC-GS (section F), SB-ECE (section 4.1), and S-AvUC (section 4.2). We also consider the setting with no secondary loss. This leads to 18 configurations ($3 \times 6$) for CIFAR-100, 18 configurations ($3 \times 6$) for CIFAR-10 and 6 configurations ($1 \times 6$) for ImageNet for a total of 42 hyperparameter tunings. The results for the "best" (section G) hyperparameter configuration for each of these settings are listed in Table 2.

In order to measure calibration of a model we consider (1) the ECE of the trained model, (2) the ECE with standard post-hoc temperature scaling, and (3) the ECE with post-hoc temperature scaling for the SB-ECE objective. The ECE measured here is $\ell_2$ norm equal-mass ECE with 15 bins. We showed in Figure 1 that the S-AvUC loss performs best aggregated across settings of datasets and primary losses. However, there isn't one secondary loss that is always the best for every setting. Even so, we do see that for all three datasets, the best ECE values for the trained model result from Soft Calibration Objectives used as secondary losses. These wins come at the cost of less than 1% accuracy. Soft Calibration Objectives are also either the best or equivalent to other methods for all datasets when we consider the numbers after temperature scaling. Note that post-hoc temperature scaling doesn't always help and should be used only if applying it results in better calibration than the trained model on some held out dataset.

## B   Train Set Memorization and Interleaved Training

The NLL primary loss has recently shown to heavily overfit the train ECE [Mukhoti et al., 2020] in some settings. In these cases, it essentially memorizes the train set, achieving near-perfect accuracy and calibration on it without help from any calibration-incentivizing losses. This raises a question about the effectiveness of using soft calibration during training for reducing test ECE in such settings.

We saw this happen on 2 of our 7 dataset + primary loss settings (CIFAR-100/10 datasets with the NLL primary loss; see table 3). As expected, using Soft Calibration Objectives as secondary training losses did not help reduce train ECE here. To fix this issue, we modified the training procedure. The train set was split into two - the 'majority' train set and the 'held-out' train set. Each epoch was also correspondingly split into two - the first part optimized NLL on the majority train set and the second part optimized Soft Calibration Objectives on the held-out train set. This way we avoided incentivizing something that was already overfit. The second part of each epoch incentivized the predicted distribution to have higher entropy. We call this 'interleaved training'.

We observed that Soft Calibration Objectives with interleaved training helped reduce test ECE relative to baseline on CIFAR-100 + NLL, as can be seen in table 2. For a fixed primary loss, the recommendation to practitioners is to either (1) have a calibration dataset and use interleaving if it yields better ECE on it or (2) in absence of a calibration dataset, use interleaving if train ECE is suspiciously low (e.g. less than 1%). Another approach is to replace the primary loss which overfits (e.g. NLL) with one that doesn't (e.g. Focal) and then use Soft Calibration Objectives for further gains.

## C   Soft Calibration Objectives for Post-Hoc Temperature Scaling

We have seen in Figure 2 that temperature scaling for the SB-ECE objective (TS-SB-ECE) outperforms standard temperature scaling (TS) both in- and out-of-distribution for models trained with the popular NLL loss on the CIFAR-100 and CIFAR-10 datasets. We see in figure 9 that this also holds true for the Focal and MSE primary losses. Whereas temperature scaling does not always help to improve calibration, particularly out-of-distribution, we do see that TS-SB-ECE outperforms standard TS in most cases. We plot comparisons for only 6 of our 42 settings in figure 9 for the sake of conciseness - these are for the CIFAR-100 and CIFAR-10 datasets for models trained using each of the three primary losses. The plots demonstrate that TS-SB-ECE outperforms TS on the i.i.d. test

Table 2: Results for the full cross-product of experimental settings between primary and secondary losses for CIFAR-100, CIFAR-10 and ImageNet. We report average accuracy, ECE and ECE obtained after post-hoc temperature scaling (TS, TS-SB-ECE) for each of the 42 configurations after hyperparameter tuning. These metrics corresponding to the best hyperparameter configuration for that setting. ECE is computed with the $\ell_2$ norm and equal-mass binning. Each cell reports average $\pm$ SEM across 10 runs. The best ECE numbers for each dataset in each of the three ECE columns are highlighted. We see that the best ECE values for all three datasets are obtained using Soft Calibration Objectives as secondary losses.

| Dataset | Primary | Secondary | Accuracy | ECE | ECE+TS | ECE+TS-SB-ECE |
|---|---|---|---|---|---|---|
| CIFAR-100 | NLL | <none> | $78.7 \pm 0.122$ | $9.10 \pm 0.139$ | $5.36 \pm 0.091$ | $4.69 \pm 0.069$ |
| | | MMCE | $77.2 \pm 0.072$ | $4.77 \pm 0.121$ | $4.06 \pm 0.138$ | $4.11 \pm 0.173$ |
| | | AvUC | $76.2 \pm 0.146$ | $11.4 \pm 0.175$ | $4.40 \pm 0.190$ | $7.64 \pm 0.187$ |
| | | AvUC-GS | $78.3 \pm 0.126$ | $6.21 \pm 0.084$ | $8.36 \pm 0.166$ | $5.40 \pm 0.241$ |
| | | S-AvUC | $78.6 \pm 0.079$ | $6.10 \pm 0.095$ | $9.13 \pm 0.085$ | $5.88 \pm 0.243$ |
| | | SB-ECE | $77.7 \pm 0.167$ | $6.86 \pm 0.839$ | $\mathbf{3.18} \pm 0.093$ | $3.51 \pm 0.314$ |
| | Focal | <none> | $78.3 \pm 0.086$ | $4.66 \pm 0.130$ | $6.47 \pm 0.140$ | $4.84 \pm 0.115$ |
| | | MMCE | $77.3 \pm 0.104$ | $5.07 \pm 0.147$ | $3.58 \pm 0.098$ | $\mathbf{2.82} \pm 0.110$ |
| | | AvUC | $77.7 \pm 0.145$ | $5.68 \pm 0.181$ | $5.87 \pm 0.182$ | $3.93 \pm 0.209$ |
| | | AvUC-GS | $78.1 \pm 0.056$ | $3.38 \pm 0.109$ | $4.89 \pm 0.118$ | $2.91 \pm 0.122$ |
| | | S-AvUC | $78.1 \pm 0.084$ | $\mathbf{1.57} \pm 0.122$ | $4.15 \pm 0.090$ | $2.89 \pm 0.077$ |
| | | SB-ECE | $78.6 \pm 0.062$ | $2.30 \pm 0.105$ | $5.16 \pm 0.108$ | $4.10 \pm 0.095$ |
| | MSE | <none> | $77.9 \pm 0.089$ | $4.17 \pm 0.122$ | $5.22 \pm 0.126$ | $4.79 \pm 0.281$ |
| | | MMCE | $78.0 \pm 0.071$ | $4.22 \pm 0.085$ | $5.06 \pm 0.108$ | $4.90 \pm 0.111$ |
| | | AvUC | $77.6 \pm 0.134$ | $4.27 \pm 0.067$ | $5.29 \pm 0.125$ | $4.42 \pm 0.224$ |
| | | AvUC-GS | $77.8 \pm 0.041$ | $4.31 \pm 0.114$ | $5.03 \pm 0.091$ | $4.15 \pm 0.135$ |
| | | S-AvUC | $77.6 \pm 0.104$ | $3.69 \pm 0.122$ | $4.60 \pm 0.144$ | $4.26 \pm 0.322$ |
| | | SB-ECE | $77.9 \pm 0.071$ | $4.14 \pm 0.100$ | $5.00 \pm 0.092$ | $4.27 \pm 0.184$ |
| CIFAR-10 | NLL | <none> | $95.5 \pm 0.040$ | $5.59 \pm 0.119$ | $1.95 \pm 0.127$ | $1.16 \pm 0.106$ |
| | | MMCE | $95.0 \pm 0.031$ | $1.55 \pm 0.053$ | $\mathbf{1.09} \pm 0.098$ | $1.45 \pm 0.114$ |
| | | AvUC | $95.5 \pm 0.053$ | $5.27 \pm 0.101$ | $2.39 \pm 0.110$ | $1.30 \pm 0.106$ |
| | | AvUC-GS | $95.6 \pm 0.036$ | $4.94 \pm 0.109$ | $2.07 \pm 0.129$ | $1.31 \pm 0.072$ |
| | | S-AvUC | $95.4 \pm 0.027$ | $5.87 \pm 0.113$ | $2.24 \pm 0.130$ | $1.20 \pm 0.115$ |
| | | SB-ECE | $95.4 \pm 0.043$ | $5.84 \pm 0.121$ | $2.28 \pm 0.091$ | $1.27 \pm 0.139$ |
| | Focal | <none> | $95.0 \pm 0.085$ | $5.45 \pm 0.079$ | $2.69 \pm 0.190$ | $1.77 \pm 0.115$ |
| | | MMCE | $95.1 \pm 0.068$ | $8.74 \pm 0.059$ | $3.13 \pm 0.110$ | $2.16 \pm 0.210$ |
| | | AvUC | $94.1 \pm 0.128$ | $1.34 \pm 0.084$ | $2.53 \pm 0.124$ | $1.12 \pm 0.133$ |
| | | AvUC-GS | $95.2 \pm 0.063$ | $1.39 \pm 0.081$ | $2.30 \pm 0.125$ | $1.46 \pm 0.108$ |
| | | S-AvUC | $94.4 \pm 0.145$ | $1.58 \pm 0.146$ | $1.34 \pm 0.172$ | $\mathbf{1.05} \pm 0.197$ |
| | | SB-ECE | $95.1 \pm 0.056$ | $\mathbf{1.19} \pm 0.088$ | $2.08 \pm 0.143$ | $1.38 \pm 0.187$ |
| | MSE | <none> | $95.0 \pm 0.041$ | $6.58 \pm 0.034$ | $5.33 \pm 0.122$ | $4.43 \pm 0.106$ |
| | | MMCE | $95.1 \pm 0.034$ | $2.82 \pm 0.046$ | $3.23 \pm 0.137$ | $2.34 \pm 0.141$ |
| | | AvUC | $94.8 \pm 0.050$ | $3.64 \pm 0.066$ | $4.72 \pm 0.139$ | $4.01 \pm 0.178$ |
| | | AvUC-GS | $94.8 \pm 0.031$ | $5.44 \pm 0.146$ | $5.56 \pm 0.151$ | $5.20 \pm 0.173$ |
| | | S-AvUC | $92.0 \pm 0.117$ | $3.40 \pm 0.126$ | $1.87 \pm 0.080$ | $1.50 \pm 0.134$ |
| | | SB-ECE | $95.1 \pm 0.056$ | $4.37 \pm 0.143$ | $5.08 \pm 0.159$ | $4.26 \pm 0.136$ |
| ImageNet | NLL | <none> | $75.8 \pm 0.036$ | $3.81 \pm 0.043$ | $2.17 \pm 0.045$ | $3.32 \pm 0.043$ |
| | | MMCE | $75.2 \pm 0.048$ | $3.12 \pm 0.025$ | $2.18 \pm 0.029$ | $2.68 \pm 0.022$ |
| | | AvUC | $75.9 \pm 0.035$ | $3.48 \pm 0.041$ | $3.37 \pm 0.037$ | $3.18 \pm 0.036$ |
| | | AvUC-GS | $75.8 \pm 0.035$ | $3.84 \pm 0.030$ | $3.15 \pm 0.028$ | $3.44 \pm 0.034$ |
| | | S-AvUC | $75.6 \pm 0.053$ | $\mathbf{2.26} \pm 0.055$ | $2.02 \pm 0.041$ | $\mathbf{1.92} \pm 0.046$ |
| | | SB-ECE | $74.7 \pm 0.028$ | $3.11 \pm 0.039$ | $\mathbf{1.92} \pm 0.024$ | $2.62 \pm 0.039$ |

Table 3: Overcofident training set memorization happens in 2 of our 7 dataset + primary loss settings. This is characterized by a very low train ECE (in bold) and a high ratio of train ECE to test ECE (in bold). ECE is computed with the $\ell_1$ norm and equal-width binning. We use interleaved training with Soft Calibration Objectives as secondary losses in these 2 cases to reduce train ECE.

| Dataset | Primary Loss | Test ECE | Train ECE | Test ECE / Train ECE |
|---------|-------------|----------|-----------|----------------------|
| CIFAR-100 | NLL | 6.88% | **0.45%** | **15.14** |
| | Focal | 4.21% | 9.18% | 0.46 |
| | MSE | 3.51% | 2.49% | 1.41 |
| CIFAR-10 | NLL | 2.66% | **0.05%** | **49.82** |
| | Focal | 5.28% | 7.13% | 0.74 |
| | MSE | 6.53% | 9.82% | 0.66 |
| Imagenet | NLL | 3.35% | 4.34% | 0.77 |

set and under all levels of skew in the CIFAR-100-C and CIFAR-10-C datasets. Table 2 shows that TS-SB-ECE outperforms TS in 35 of the 42 settings that we experiment on. We conclude that Soft Calibration Objectives can be used to improve upon TS - the popular post-hoc calibration method.

A pertinent follow-up question for temperature scaling is whether we can take this approach of directly optimizing temperature for SB-ECE a step further and directly optimize temperature for (hard-binned) ECE instead. ECE is not trainable, but we might still be able to do a search for temperature. Indeed, multi-resolution search to optimize temperature for hard-binned ECE (hereby, TS-HB-ECE) is an alternative to gradient-based training of temperature for soft-binned ECE (i.e. TS-SB-ECE). Our results suggest that this might be promising. This approach is worth investigating further.

The two methods have potential advantages and disadvantages. TS-HB-ECE has higher variance than TS-SB-ECE for a given sample size, which risks a suboptimal solution even if TS-HB-ECE is the quantity we wish to optimize. TS-HB-ECE is also less computationally efficient than TS-SB-ECE. To formally compare complexities, assume that we want to compute top-label equal-width ECE and that we save logits in the last training epoch. Let $K$ be the number of classes, $N_c$ be size of the recalibration dataset and $V$ be the number of temperature values inspected for multi-resolution search. Assuming that the gradient-based method trains for a small constant number of epochs, the post-processing complexity for TS-SB-ECE is $O(N_c K)$ and for TS-HB-ECE is $O(V N_c K)$. The quantity $V$ may not be small if we want to get close to the optimal temperature. Consequently, this cost difference can be high for language models with large vocabularies and a lot of training data.

# D  Reliability Plots

In this section, we use reliability plots (figure 10) to compare the calibration of models trained using different training objectives for CIFAR-100, CIFAR-10 and ImageNet. We compare the cross-entropy baseline with each of the proposed methods to train for calibration: Focal loss [Lin et al., 2018], MSE Loss [Hui and Belkin, 2021], MMCE loss [Kumar et al., 2018], S-AvUC and SB-ECE. We do not include the AvUC [Krishnan and Tickoo, 2020] loss since it does not help to improve calibration for non-Bayesian neural networks (section F). We find that NLL results in overconfident models and that adding MMCE as a secondary loss to the NLL primary loss reduces the amount of overconfidence. The MSE primary loss results in models that are overconfident for some confidence bins and underconfident for others, which helps to explain why temperature scaling is not as effective for MSE as compared to Focal and NLL (see table 2). Focal loss, on the other hand, results in underconfident models. This is also seen in the ECE vs average uncertainty plots (figure 10), where we find that NLL results in least average uncertainty for all datasets whereas Focal loss is amongst the highest average uncertainties for both CIFAR-100 and CIFAR-10. Obtaining both lower ECE and lower average uncertainty simultaneously as compared to the standard cross-entropy loss remains an open challenge. Finally we note that models trained using Soft Calibration Objectives as secondary losses are the most visually calibrated for all three datasets, consistent with our findings in Section 5.

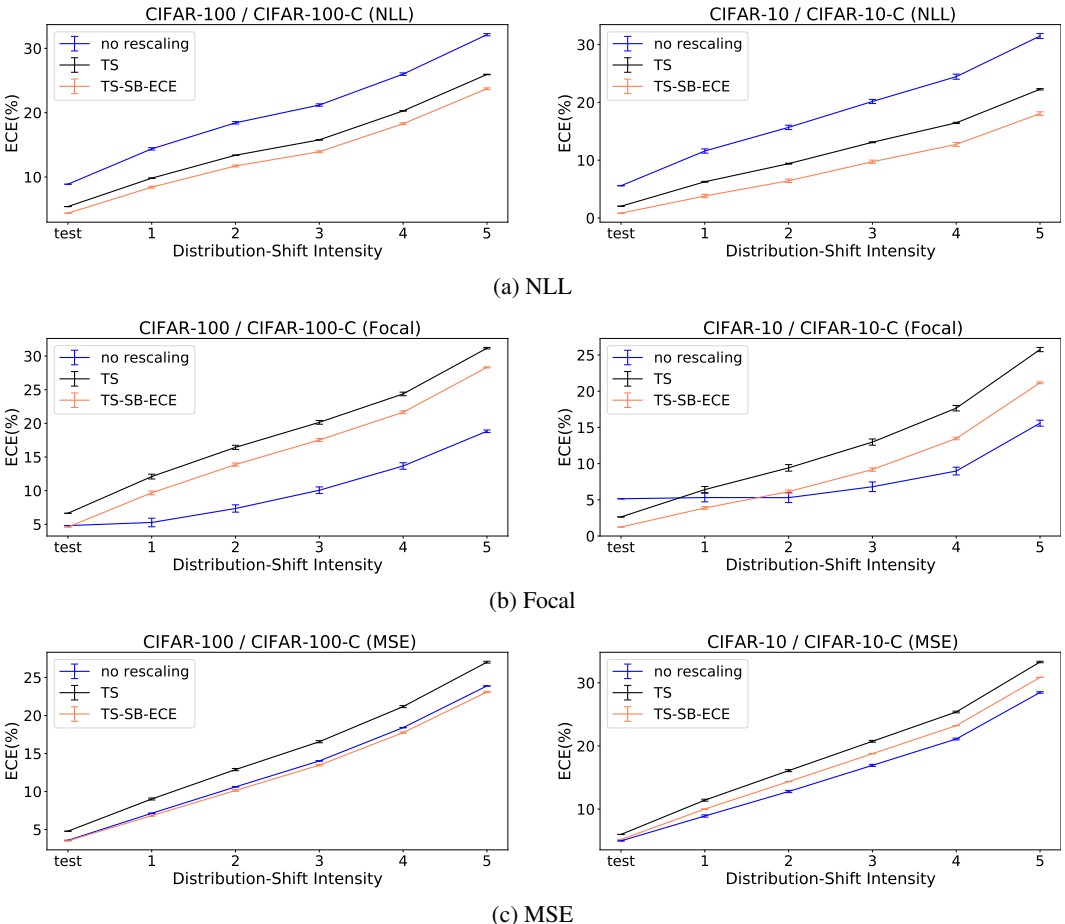

(a) NLL

(b) Focal

(c) MSE

Figure 9: Post-hoc temperature scaling with the soft calibration error objective (TS-SB-ECE) outperforms standard post-hoc temperature scaling (TS), particularly under distribution shift. This result holds across datasets (left and right panels), distribution shift intensities (along abscissa) and training objectives (rows). The ECE value (equal-mass binning, $\ell_2$ norm) shown is the mean ECE across all corruption types in CIFAR-10-C and CIFAR-100-C. Error bars are $\pm 1$ standard error of mean (SEM), corrected for intrinsic variability due to type of corruption [Masson and Loftus, 2003]
.

# E   Other Calibration Measures

Even though ECE is the most popular metric for measuring calibration, it has issues related to consistency and bias [Nixon et al., 2019, Kumar et al., 2019, Roelofs et al., 2020, Gupta et al., 2020]. Debiased CE [Kumar et al., 2019] and mean-sweep CE [Roelofs et al., 2020] have been shown to have lesser bias and more consistency across the number of bins parameter than ECE whereas KS-error [Gupta et al., 2020] avoids binning altogether.

We have validated our findings on CIFAR-100 and CIFAR-10 using KS-error and mean-sweep CE. The findings based on these measures (table 4) are consistent with those based on ECE i.e. soft-calibration objectives outperform all other methods. We have reported results corresponding only to tables 1a and 1b for conciseness rather than those correponding to the full table 2. As stated before, these account for the best-performing experiments on these datasets and in particular the best results for the cross-entropy baseline, Focal loss [Mukhoti et al., 2020] and MMCE [Kumar et al., 2018].

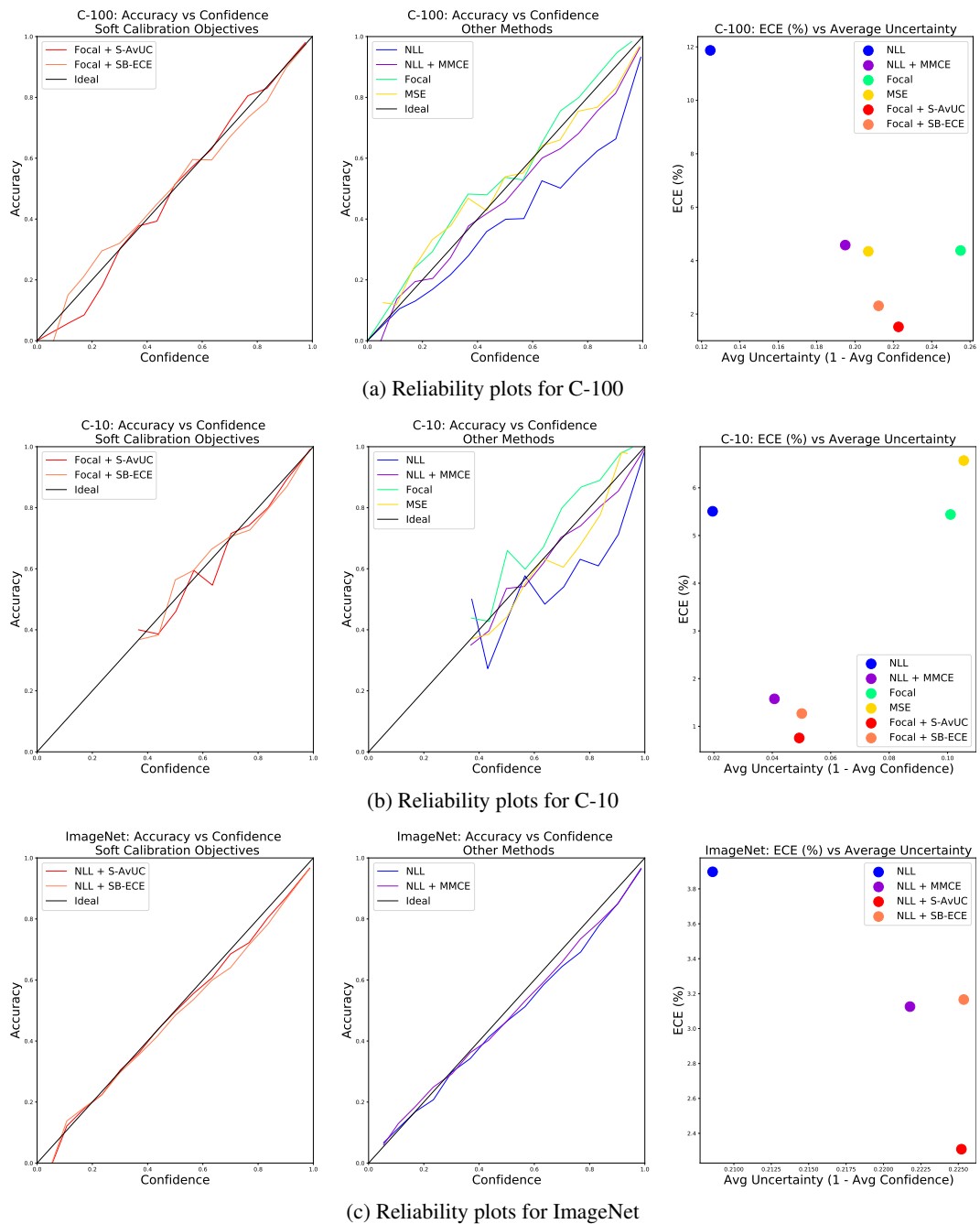

(a) Reliability plots for C-100

(b) Reliability plots for C-10

(c) Reliability plots for ImageNet

Figure 10: Models trained with the NLL loss are overconfident across all three datasets. Adding MMCE as a secondary loss reduces the overconfidence. The MSE loss results in underconfidence and overconfidence for different bins. Models trained with Focal loss are underconfident. Soft Calibration Objectives (S-AvUC, SB-ECE) result in the most visually calibrated reliability plots across all datasets. Note that the high confidence regions have much higher density than the low confidence regions and are thus more critical to the ECE value.

In the ECE vs average uncertainty plots, we see that Focal and NLL losses result in the highest and lowest average uncertainties respectively. S-AvUC results in the lowest ECE in all three datasets and SB-ECE is the next best for CIFAR-10 and CIFAR-100. Obtaining lower ECE as well as lower average uncertainty as compared to the NLL loss remains an open challenge.

Table 4: We report accuracy, average ECE, KS-Error and mean-sweep CE for the CIFAR-10 and CIFAR-100 datasets corresponding to the entries in tables 1a and 1b. Mean-sweep CE is computed with the $\ell_2$ norm and equal-mass binning. KS-error is computed as originally defined with the $\ell_1$ norm. These additional measures further demonstrate that soft-calibration objectives outperform all other methods.

(a) CIFAR-100

| Loss Fn. | Accuracy | ECE | KS-Error | mean-sweep CE |
|---|---|---|---|---|
| NLL | 78.7 | 9.10 | 6.71 | 9.02 |
| NLL + MMCE | 77.2 | 4.77 | 3.27 | 4.77 |
| Focal | 78.3 | 4.66 | 4.20 | 4.57 |
| Focal + SB-ECE | 78.6 | 2.30 | 0.71 | 2.21 |
| Focal + S-AvUC | 78.1 | **1.57** | **0.51** | **1.24** |

(b) CIFAR-10

| Loss Fn. | Accuracy | ECE | KS-Error | mean-sweep CE |
|---|---|---|---|---|
| NLL | 95.5 | 5.59 | 2.64 | 4.78 |
| NLL + MMCE | 95.0 | 1.55 | 0.83 | 1.30 |
| Focal | 95.0 | 5.45 | 5.22 | 5.41 |
| Focal + SB-ECE | 95.1 | **1.19** | **0.41** | **0.77** |
| Focal + S-AvUC | 94.4 | 1.58 | 0.66 | 1.43 |

# F    AvUC with Gradient Stopping

The AvUC loss [Krishnan and Tickoo, 2020] was proposed to train for calibration in Stochastic Variational Inference (SVI) settings. It is based on the idea of giving an incentive to the model to be certain when accurate and uncertain when inaccurate via a secondary loss. The secondary loss term is described in equations 13 and 14. These are restated here for readability.

$$\text{AvUC}(\kappa, \hat{D}, \boldsymbol{\theta}) = \log\left(1 + \frac{n_{\text{AU}} + n_{\text{IC}}}{n_{\text{AC}} + n_{\text{IU}}}\right), \tag{13}$$

$$\begin{aligned} n_{\text{AU}} &= \Sigma_{i|(\boldsymbol{x}_i,y_i)\in S_{\text{AU}}}(c_i \tanh h_i) & n_{\text{IC}} &= \Sigma_{i|(\boldsymbol{x}_i,y_i)\in S_{\text{IC}}}((1-c_i)(1-\tanh h_i)) \\ n_{\text{AC}} &= \Sigma_{i|(\boldsymbol{x}_i,y_i)\in S_{\text{AC}}}(c_i(1-\tanh h_i)) & n_{\text{IU}} &= \Sigma_{i|(\boldsymbol{x}_i,y_i)\in S_{\text{IU}}}((1-c_i)\tanh h_i). \end{aligned} \tag{14}$$

Note that $S_{AU}$, $S_{IC}$, $S_{AC}$ and $S_{IU}$ form a partition of datapoints from the training batch $\hat{D}$ which fall in each of the four categories resulting from two classifications: (1) accurate [A] vs. inaccurate [I] based on whether the model's prediction is correct and (2) certain [C] vs uncertain [U] whether the model's entropy is above or below a threshold $\kappa$. In our experiments we tune $\kappa$ rather than inferring it from the first few epochs as was suggested in [Krishnan and Tickoo, 2020]. Despite this additional degree of freedom, we consistently observe that the originally proposed AvUC loss does not help for calibration in non-Bayesian settings.

A closer look at equations 13 and 14 suggests that this might be because of some of the incentives provided by the secondary loss. As observed in section 4.2, minimizing the AvUC loss results in the model being incentivized to be even more confident in its inaccurate and certain predictions via minimizing $n_{\text{IC}}$, specifically the $(1-c_i)$ multiplicand. Similarly, it also encourages the model to be even less confident in its accurate and uncertain predictions via minimizing $n_{\text{AU}}$, specifically the $c_i$ multiplicand.

To test whether these misincentives are really the cause of our observations, we conduct experiments with a variant of the loss where we stop gradients flowing through the $(1-c_i)$ and $c_i$ multiplicands in each of the four expressions in equation 14. We denote this modified version of the AvUC loss as the AvUC-GS loss, where "GS" denotes gradient stopping. The experiments confirm our hypothesis - we observe that the AvUC-GS secondary loss helps in improving calibration in situations where the original AvUC secondary loss did not. This observation holds across datasets and primary losses. This

can be seen in table 2, where the rows corresponding to AvUC-GS in the secondary loss column have lower ECE on an average than the respective rows corresponding to the AvUC secondary loss. We conclude that AvUC-GS is also an effective secondary loss. However, the soft calibration objective S-AvUC which is inspired from AvUC-GS generalizes, outperforms and supersedes it.

## G   Hyperparameter Tuning and the Accuracy-Calibration Tradeoff

We look at both the accuracy and the ECE of competing hyperparameter configurations. Some comparisons yield a clear winner but often there is a tradeoff between these. In such cases, we choose the lowest ECE whilst giving up less than 1% accuracy relative to the hyperparameter configuration with the highest accuracy.

As stated in section 5.1, we use the Wide-Resnet-28-10 architecture [Zagoruyko and Komodakis, 2017] trained for 200 epochs on CIFAR-100 and CIFAR-10. For Imagenet, we use the Resnet-50 [He et al., 2015] architecture trained for 90 epochs. The loss function we use in our experiments is $PL + \beta \cdot SL + \lambda \cdot L2$ where PL and SL denote the primary and secondary losses respectively and L2 denotes the weight normalization term with $\ell_2$ norm.

All our experiments use the SGD with momentum optimizer with momentum fixed to 0.9 and base learning rate fixed to 0.1. We follow a learning rate schedule which is fixed for each dataset across training losses. The number of bins ($M$, if applicable) for the soft binning secondary loss is fixed to 15. We used a per-core-batch-size of 64 on a 2x2 TPU topology with 8 cores for an effective batch size of 512. Both the baseline runs and experimental runs (all runs from Table 2) used this batch size. In general, larger batch size is better for the computation of soft calibration losses, but we did not see significant ECE gains with a further increase in batch size. These form the set of hyperparameters we fixed rather than tuned. Fixing some hyperparamers allows us to keep the search space manageable. The values we use for these are those which work well for the cross-entropy baseline.

The $\beta$ (if applicable) and $\lambda$ parameters along with parameters relevant to the secondary loss are the set of hyperparameters that we tune. In our experiments with the SB-ECE secondary loss, the secondary loss function we use is SB-ECE$_{lb,p}(M, T, \hat{D}, \boldsymbol{\theta})$ from equation 12, for which we tune the $T$ parameter. In our experiments with the Soft-AvUC loss the secondary loss function we use is S-AvUC$(\kappa, T, \hat{D}, \boldsymbol{\theta})$ from equation 15, for which we tune the $\kappa$ and $T$ parameters. As stated above, these combined with $\beta$ and $\lambda$ form our set of tuned parameters. These hyperparameters are the most critical ones for demonstrating the effectiveness of our techniques.

We tune these parameters one-at-a-time starting with the threshold parameter for the secondary loss ($\kappa$, if applicable), followed by temperature parameter for the secondary loss ($T$, if applicable), the $\beta$ parameter (if applicable) and the L2-normalization coefficient $\lambda$. We retain the best value from the tuning experiments for one parameter while tuning a subsequent parameter.

We show in table 2 that Soft Calibration Objectives result in lower ECEs than previous methods in exchange for a small reduction in accuracy relative to the cross-entropy baseline. In our experiments, we found that other methods to train for calibration (MMCE, AvUC, Focal loss, MSE loss) also have to sacrifice a small amount of accuracy relative to the cross-entropy baseline in order to attain better calibrated models. This fundamental tradeoff can be summarized by the pareto-optimal curve between accuracy and calibration. Our methods result in points on this curve which are better calibrated than previously proposed methods, whilst trading off less than 1% accuracy.

## H   Is SB-ECE as secondary loss a proper scoring rule?

We start this discussion by asking the following question: what happens to SB-ECE for perfectly calibrated models as the dataset size goes to infinity? We know that ECE tends to zero in this case. Proposition 1 shows that the same holds for SB-ECE. We will use terminology introduced in sections 3 and 4 in this section.

**Proposition 1.** *Consider a dataset $D = \langle (\boldsymbol{x}_i, y_i) \rangle_{i=1}^{N}$ drawn from the joint probability distribution $\mathcal{D}(\mathcal{X}, \mathcal{Y})$. Say we have a perfectly calibrated model for it such that $E[a|c] = c$, where $a$ and $c$ denote accuracy and confidence respectively. If we consider SB-ECE$_{bin,p}$ with $M$ bins, temperature $T$ and norm $p$ as defined in equation 11, we have*

$$SB\text{-}ECE_{bin,p}(M, T, \hat{D}, \boldsymbol{\theta}) \rightarrow 0 \text{ as } N \rightarrow \infty$$

*Proof.* Let $p_c$ denote the p.d.f. of the confidence $c$ viewed as a random variable. Let us denote the membership function for bin $j$ by $u_j(c)$, a shorthand for our earlier notation $u_{M,T,j}(c)$ from section 4.1. The size, confidence and accuracy of bin $j$, as defined in equations 9 and 10 are denoted by $C_j$ and $A_j$ respectively. We observe that as $N \to \infty$, these quantities satisfy the following:

$$C_j = \frac{\Sigma_{i=1}^{N} u_j(c_i) \cdot c_i}{\Sigma_{i=1}^{N} u_j(c_i)} \to \frac{\int_0^1 x u_j(x) \, p_c(x) dx}{\int_0^1 u_j(x) \, p_c(x) dx}$$

$$A_j = \frac{\Sigma_{i=1}^{N} u_j(c_i) \cdot a_i}{\Sigma_{i=1}^{N} u_j(c_i)} \to \frac{\int_0^1 E[a|c=x] u_j(x) \, p_c(x) dx}{\int_0^1 u_j(x) \, p_c(x) dx}$$

Since $E[a|c=x] = x$ for perfectly calibrated models, we infer that as $N \to \infty$:

$$C_j - A_j \to 0$$

$$\text{SB-ECE}_{\text{bin},p}(M, T, \hat{D}, \boldsymbol{\theta}) \to 0$$

$\square$

Note that this does not necessarily hold for datasets of a given finite size. If we measure SB-ECE with $N = 2$ datapoints for a perfectly-calibrated model which always has a confidence of $30\%$ and is correct $30\%$ of the time, we will always end up with ECE and SB-ECE both greater than zero.

If we consider an optimal classifier which always outputs the true probabilities, then it minimizes NLL since NLL is a proper scoring rule and it minimizes SB-ECE (note that SB-ECE $\geq 0$ by definition) as dataset size goes to infinity as per proposition 1 since it is perfectly calibrated. Hence, it minimizes any positive linear combination of NLL and SB-ECE which implies that these linear combinations are proper scoring rules in the limit of infinite data.

# I   Post-hoc Dirichlet Calibration

In Tables 1 and 2, we have compared our methods to existing calibration-incentivizing losses that operate during training, with and without post-hoc temperature scaling. There are a several post-hoc recalibration techniques which can be used complementary to our methods. For this reason, comparing to these has not been the focus of our work, similar to [Mukhoti et al., 2020] and [Kumar et al., 2018]. Nevertheless, in this section we show that our methods do significantly better than training with the cross-entropy, focal or MSE primary losses and using such post-hoc recalibration methods. In particular, we compare against Dirichlet calibration [Kull et al., 2019]. Table 5 records ECE numbers for the best-performing training objectives for CIFAR-10 corresponding to table 1b, both with and without post-hoc Dirichlet calibration.

Table 5: We report accuracy and average ECE across 10 runs for the best performing training objectives for the CIFAR-10 dataset corresponding to the entries in table 1b, both with and without post-hoc dirichlet calibration. ECE is computed with the $\ell_2$ norm and equal-mass binning. Our methods outperform post-hoc dirichlet calibration applied to the NLL, Focal and MSE primary losses.

(a) CIFAR-10

| Loss Fn. | Accuracy | ECE | ECE w/ Dirichlet |
|---|---|---|---|
| NLL | 95.5 | 5.59 | 2.45 |
| NLL + MMCE | 95.0 | 1.55 | **1.28** |
| Focal | 95.0 | 5.45 | 2.48 |
| Focal + SB-ECE | 95.1 | **1.19** | 2.24 |
| Focal + S-AvUC | 94.4 | 1.58 | 1.70 |
| MSE | 95.0 | 6.58 | 4.69 |

# J  Multiclass Calibration Error

Reducing top-label calibration error using calibration-incentivizing training loss functions is the focus of our work, similar to [Kumar et al., 2018], [Krishnan and Tickoo, 2020] and [Mukhoti et al., 2020]. In this section, we additionally evaluate marginal ECE [Kumar et al., 2019] which measures the calibration of the predicted probability distribution over all classes rather than just the top class. Table 6 shows that our methods which are trained to minimize top-label ECE yield the best marginal ECE numbers as well.

The terms in the S-AvUC loss are tied to the top-label and this does readily lend itself to a multiclass extension. The soft binning approach however immediately yields a trainable soft version of marginal ECE analogous to the top-label case. Soft-binned marginal ECE is beyond the scope of this paper but can be investigated further.

Table 6: We report accuracy, average top-label ECE and average marginal ECE for the CIFAR-10 and CIFAR-100 datasets corresponding to the entries in tables 1a and 1b. Marginal ECE and top-label ECE are computed with the $\ell_2$ norm and equal-mass binning. Soft-calibration objectives outperform other methods on multiclass CE in addition to the top-label CE they are designed to optimize.

(a) CIFAR-100

| Loss Fn. | Accuracy | ECE | Marginal ECE |
|---|---|---|---|
| NLL | 78.7 | 9.10 | 3.78 |
| NLL + MMCE | 77.2 | 4.77 | 3.05 |
| Focal | 78.3 | 4.66 | 4.35 |
| Focal + SB-ECE | 78.6 | 2.30 | 3.05 |
| Focal + S-AvUC | 78.1 | **1.57** | **2.72** |

(b) CIFAR-10

| Loss Fn. | Accuracy | ECE | Marginal ECE |
|---|---|---|---|
| NLL | 95.5 | 5.59 | 2.31 |
| NLL + MMCE | 95.0 | 1.55 | 2.06 |
| Focal | 95.0 | 5.45 | 5.64 |
| Focal + SB-ECE | 95.1 | **1.19** | 2.24 |
| Focal + S-AvUC | 94.4 | 1.58 | **1.65** |