# OpenReview forum: "Soft Calibration Objectives for Neural Networks"
_NeurIPS.cc/2021/Conference — NeurIPS 2021 Poster_

### Official Review · Reviewer_rMMu · 2021-07-13

**Rating:** 6
**Confidence:** 4

**Summary:**

The paper presents differentiable loss functions to improve the calibration of deep nets class probability estimates. To achieve this, the authors propose a soft approximation of binning operation, which is at the core of popular measures for model calibration error. Empirical evidence shows that the new approach achieves improved calibration performance compared to existing methods while maintaining accuracy to the extent possible.

**Limitations And Societal Impact:**

The limitations are partially described. See "main concern" above.

**Main Review:**

Originality

The proposal of a loss that improves calibration is not new. For instance, R. Krishnan and O. Tickoo took this approach in their work titled “Improving model calibration with accuracy versus uncertainty optimization.” With that said, the novelty of new resides in the explicit minimization of metrics for calibration errors (based on binning), which seems to improve performance. This approach is closer to that of “Calibration of neural networks using splines” by Gupta et al. The authors also soften the AvUC loss offered by Krishnan and Tickoo, allowing the training of non-Bayesian neural networks.

Quality

I believe the writing of the paper can be improved. There is extensive use of acronyms and notations, which make the paper hard to read and follow. I find the proposed idea straightforward; I was hoping to see a theoretical analysis for the overall approximation error obtained by the soft operations. Also, suppose we are given a perfectly calibrated model---would the SB-ECE loss will be equal to zero?

Main concern

In the experiments, the best performing combination of “accuracy loss” + “calibration loss” is not consistent across the benchmark data sets. As a result, it is not clear what should be the default approach. I also do not understand why the calibration loss reduces predictive performance; it seems to be a reasonable regularization. Would the ideal model satisfy this property? In other words, does the optimal classifier minimize the proposed losses (with infinite data)?

Significance

The problem of calibrating the probability estimates obtained by machine learning algorithms is important. The paper presents a new way to encourage calibrated predictions during training. The experiments demonstrate the effectiveness of this approach; however, it is not clear which loss works best.


**Time Spent Reviewing:**

5

---

> ### Author Response · Authors · 2021-08-11
> **Response to reviewer rMMu**
>
> Thanks for your endorsement, probing questions and insights. We have addressed your concerns and this really helped us improve the paper and indeed realize that NLL + SB-ECE is a proper scoring rule.
>
> 1. **Is the SB-ECE loss zero for perfectly calibrated models? Does the optimal classifier minimize the proposed losses (with infinite data)?**
> Thanks for this insightful question. Indeed we now note that the SB-ECE loss as stated in equation (11) of the main paper is zero for perfectly calibrated models with infinite data. We have both verified this empirically and included a simple proof in the appendix. Since the SB-ECE loss is non-negative by definition, we infer that every perfectly calibrated model minimizes SB-ECE. If the optimal classifier is the one which always outputs the true probabilities, then it minimizes NLL since NLL is a proper scoring rule and it minimizes SB-ECE since it is perfectly calibrated. Hence, it minimizes any positive linear combination of NLL and SB-ECE, which implies that these linear combinations are proper scoring rules (may not be strictly proper scoring rules). We don’t make such a claim for S-AvUC and we note that no such claim was made for the original AvUC loss *[Krishnan and Tickoo, 2020]*.
>   We also note that hard-binned ECE is a biased estimator of true ECE (see *[Roloefs et al., 2020, Kumar et al., 2019]*). Because SB-ECE includes hard-binned ECE as a special case as the temperature goes to zero, we infer that SB-ECE is also a biased estimator of true ECE.
> 2. **Not clear which loss works best. What should be the default approach?**
> This is an insightful and valid concern, and one which we worked on post-submission. Going beyond specific dataset + primary loss settings, we have run evaluations so as to be able to recommend one secondary loss across all the settings. We do this by measuring the effect size (Cohen’s d-value *[Wiki. Cohen D]*) of each secondary loss for reducing ECE across datasets and primary losses. We observe that S-AvUC results in the most effective ECE reduction both without temperature scaling (d=7.47) and with our recommended temperature scaling variant (d=4.86). Note that d>2.0 is considered a huge effect *[Wiki. Cohen D]*. We can thus recommend the use of S-AvUC as secondary loss as the default approach.
>   | (Cohen’s d-value) | Without TS-SB-ECE | With TS-SB-ECE |
> |:-----------------:|:-----------------:|:--------------:|
> |        MMCE       |        6.26       |      3.19      |
> |        AvUC       |        3.87       |      1.27      |
> |       SB-ECE      |        5.10       |      2.43      |
> |       S-AvUC      |        7.47       |      4.86      |
> 3. **Why do proposed losses reduce predictive performance?**
> In general, addition of a new secondary loss term which does not directly incentivize accuracy is expected to reduce predictive performance. Consistent with this claim, other methods with secondary losses (MMCE *[Kumar et al., 2018]*, AvUC *[Krishnan and Tickoo, 2020]*) also result in a decrease in predictive performance in our experiments.
>
> References:
>   See response to reviewer 2pg6

---

> > ### Comment · Reviewer_rMMu · 2021-08-26
> > **Follow-up**
> >
> > Thank you for the detailed response, which addressed my concerns. In particular, it is reassuring that NLL + SB-ECE is a sensible loss function. Also, thank you for clarifying that S-AvUC is most effective in terms of ECE reduction, although it is not clear whether S-AvUC is a proper loss.

---

> > > ### Author Response · Authors · 2021-08-26
> > > **Thanks**
> > >
> > > Thanks for your review and for the follow-up response. This helped us improve the paper greatly.

---

### Official Review · Reviewer_CL78 · 2021-07-14

**Rating:** 6
**Confidence:** 4

**Summary:**

This paper addresses neural networks miscalibration by introducing a soft version of two different calibration measures, namely ECE and AvUC.

To make ECE differentiable, the authors propose to soften the bin membership in ECE calculation and define differentiable bin size, bin confidence, and bin accuracy based on the soft membership function. The membership function is characterized by a categorical distribution (implemented by softmax) over the temperature-scaled Euclidean distances between the confidence score of a sample and the centers of the bins. The centers of the bins are pre-determined as the center of equal-width binning with some pre-known number of bins. The authors use the differentiable ECE as an additional term added to traditional loss functions for training classifiers. They also use this loss alone for post-hoc calibration. In addition to ECE, they soften the newly introduced AvUC loss with similar tricks.

Experiments on benchmark datasets are conducted to demonstrate the effectiveness of the proposed method regarding ECE performance metrics.

**Limitations And Societal Impact:**

YES

**Main Review:**

Overall, the formulation that leads to soft bin membership is exciting and novel.
While I appreciate the rigor that the authors aim at providing to the problem, I have several concerns/questions that need to be addressed:

- It is not mentioned in the paper how the quantities in equations 8-10 are evaluated for each sample during training. Is it required to compute the bin size, accuracy, and confidence after each training iteration on all training data?

- Post-hoc calibration using SB-ECE: I believe there is no need to optimize the soft-binned ECE to find a good temperature value. One can perform a search (multi-resolution) to find the optimal temperature using the hard-binned ECE. It is easy to evaluate the ECE for an input value of t, and it won't take too much time (compared to training the original model) to search for a good temperature value. It also keeps the objective exact without introducing the inaccuracies added by soft-binned ECE.

- In figure 1, For CIFAR datasets, SB-ECE and S-AvUC, when added to NLL, seem not to make much improvement to calibration while making accuracy worse. Adding focal loss seems to improve them. However, this seems like a mystery in the paper. It is the addition of focal loss that makes them work. Can the authors elaborate more on that and explain why this is happening?

- When training for NLL, we usually have the accuracy on the training set is 100%, and the confidence of the training samples are high, meaning that training on the NLL will result in calibrated confidence scores on training data. I am wondering how the SC-ECE and S-AvUC avoid that? Mainly, is S-AvUC a proper loss such that one can theoretically guarantee upon training one can get calibrated confidences on the training data itself? What happens if one uses these loss functions to continue training on a pre-trained model?

- The numbers in Table 1 show different accuracies/ECEs compared to a similar model in [1] (Table F.1 and Table 2).
Why are there so many differences between these measures and the numbers in Table 1 of the paper? Some details might have been missed in training (e.g., training for 350 epochs rather than 200 epochs).

- Only top-label ECE is computed. Also, only the TS method is compared. The authors have cited many strong post-hoc calibration methods in the related work but have not compared them with their method.
Furthermore, there are many cases (especially in the applications mentioned in the introduction, such as medical diagnosis)  the calibration of other classes is necessary (That is why measures like classwise calibration or marginal calibration errors have been introduced[2,3]). These are not evaluated in the paper. Also, it would be nice if the extension of soft-calibration losses is discussed for these cases. ECE itself has many flaws as one has to select the number of bins beforehand. There also biases due to small number of bins, etc., [3]. The method introduced in the paper suffers from the same issues.

- Figure S1 in the supplementary shows that the uncalibrated model is more robust under distribution shift than the proposed post-hoc method in most of the cases. These results are hidden in the main paper and have not been discussed. I hope the authors can explain the reason.

Minor:
- In abstract line 10. 82% reduction refers to which experiment/table in the main paper?
- In eq. 6 and 11, "i" in the sum should be replaced with "j."
- In eq. 14, $S_{\rm AU}$, etc., are not defined.


References:

[1] Mukhoti et al., "Calibrating Deep Neural Networks using Focal Loss," NeurIPS 2020 (arXiv:2002.09437).

[2] Kull et al., "Beyond temperature scaling: Obtaining well-calibrated multi-class probabilities with Dirichletcalibration," NeurIPS 2019.

[3] Kumar et al., "Verified Uncertainty Calibration," NeurIPS 2019.

**Time Spent Reviewing:**

10

---

> ### Author Response · Authors · 2021-08-11
> **Response to Reviewer CL78**
>
> Thanks for a detailed and insightful review. We have addressed your comments and updated the paper. Your questions also helped us realize that NLL + SB-ECE is a proper scoring rule
>
> 1. **Is bin size, confidence and accuracy computed per epoch?** We should have stated this explicitly. We compute these quantities per-training-batch. So, it is not required to compute bin sizes, accuracies and confidences after each iteration on all training data.
> 2. **Multi-resolution search for temperature (T) rather than training for soft ECE?** This is an excellent observation: multi-resolution search to optimize T for hard-binned ECE (TS-HB-ECE) is an alternative to gradient-based training with soft-binned ECE (TS-SB-ECE). The two methods have potential advantages and disadvantages. TS-HB-ECE has higher variance than TS-SB-ECE for a given sample size, which risks a suboptimal solution even if TS-HB-ECE is the quantity we wish to optimize. TS-HB-ECE is also less computationally efficient than TS-SB-ECE.
>   The comparative complexity analysis is as follows: Say we want to compute top-label equal-width ECE and that we save logits in the last training epoch. Let D be (# classes), N be (# recalibration examples) and K be (# temp values for multi-resolution search). The post-processing complexity for TS-SB-ECE is O(ND) and for TS-HB-ECE is O(NKD). This cost difference can be high for language models with large vocabularies and a lot of training data.
>   The experiment is worth conducting. Regardless, we wish to emphasize that we have introduced a method that is a reliable and significant improvement over the originally-stated *[Guo et al., 2017]* and commonly practiced *[AWS TS Presc.]* temp. scaling which trains T using a cross-entropy loss.
>   Finally, we note that a similar observation was also made in *[Krishnan and Tickoo, 2020]* which “proposes post-hoc uncertainty calibration AvUTS by extending the temp. scaling methodology to optimize the AvUC loss instead of NLL”. We have added a section to the appendix clarifying this comparison of TS-SB-ECE to TS-HB-ECE.
> 3. .
>     1. **Why do proposed objectives not work with NLL on CIFAR?** A closer look at the test/train ECE for our 7 dataset + primary loss settings (see table below with L1 norm ECE) reveals that CIFAR-100/10 + NLL are the only two cases where ECE heavily overfits the train set, consistent with observations in *[Mukhoti et al., 2020]*. Nevertheless, we note that on CIFAR-100 + NLL, soft calibration objectives reduce test ECE (table S1 in appendix) using “interleaved training” (described below in bullet point 4.1), implying that soft calibration objectives improve calibration even when train ECE overfitting is observed.
>     | (Dataset + Loss) | Test ECE | Train ECE | Test ECE / Train ECE |
> |:-------------------------------:|:--------:|:---------:|:--------------------:|
> |         CIFAR-100 + NLL         |   6.88%  |   0.45%   |         15.14        |
> |        CIFAR-100 + Focal        |   4.21%  |   9.18%   |         0.46         |
> |         CIFAR-100 + MSE         |   3.51%  |   2.49%   |         1.41         |
> |          CIFAR-10 + NLL         |   2.66%  |   0.05%   |         49.82        |
> |         CIFAR-10 + Focal        |   5.28%  |   7.13%   |         0.74         |
> |          CIFAR-10 + MSE         |   6.53%  |   9.82%   |         0.66         |
> |          Imagenet + NLL         |   3.35%  |   4.34%   |         0.77         |
>     2. **Is it Focal loss which makes proposed objectives work?** Going beyond specific explanations for individual dataset + primary loss settings, we have run evaluations so as to be able to recommend one secondary loss across all the settings. We do this by measuring the effect size (Cohen’s d-value *[Wiki. Cohen D]*) of each secondary loss for reducing ECE across datasets and primary losses (see table below). We observe that S-AvUC results in the most effective ECE reduction both without temp. scaling (d=7.47) and with our recommended temp. scaling (d=4.86). Note that d>2.0 is considered a huge effect *[Wiki. Cohen D]*
>   | (Cohen’s d-value) | Without TS-SB-ECE | With TS-SB-ECE |
> |:-----------------:|:-----------------:|:--------------:|
> |        MMCE       |        6.26       |      3.19      |
> |        AvUC       |        3.87       |      1.27      |
> |       SB-ECE      |        5.10       |      2.43      |
> |       S-AvUC      |        7.47       |      4.86      |
> 4.
>     1. **How do you avoid very high confidence on the Train set, known problem with NLL?** This is an excellent insight and one which we have paid close attention to in our experiments. We do observe in 2 out of our 7 dataset + primary loss settings (CIFAR-100/10 + NLL) that train ECE is overfit (see table in bullet point 3.1) and that reducing it further does not help. For these two cases only, our hypothesis was that this could be fixed by doing “interleaved training”. Each epoch is split into two - the first part optimizes NLL on the majority train set and the second part optimizes soft calibration objectives on the “held out” train set. This way we avoid fitting to something that is already overfit. The second part of each epoch incentivizes the predicted distribution to have higher entropy. The hypothesis was validated: soft calibration objectives with interleaved training help reduce test ECE relative to the NLL baseline on CIFAR-100 as can be seen in Table S1. We have now clarified this in the appendix.
>     2. **Are these proper scoring rules?** SB-ECE loss as stated in equation (11) of the main paper is zero for perfectly calibrated models with infinite data. We have both verified this empirically and included a simple proof in the appendix. Since the SB-ECE loss is non-negative by definition, we infer that every perfectly calibrated model minimizes SB-ECE. If the optimal classifier is the one which always outputs the true probabilities, then it minimizes NLL since NLL is a proper scoring rule and it minimizes SB-ECE since it is perfectly calibrated. Hence, it minimizes any positive linear combination of NLL and SB-ECE which implies that these linear combinations are proper scoring rules (may not be strictly proper scoring rules). We don’t make such a claim for S-AvUC and we note that no such claim was made for the original AvUC loss *[Krishnan and Tickoo, 2020]*.
> 5. **Differences relative to Focal loss paper?** First, we note that the accuracies are comparable. The baseline NLL accuracy we report for CIFAR-100 is 78.7% (Wide-Resnet-28-10), as opposed to 79.3% (Wide-Resnet-26-10) in *[Mukhoti et al., 2020]*. *[Kumar et al., 2019]* reports 74.0% for the same setting (Wide-Resnet-28-10). The main discrepancy between ECE can be explained by the fact that we compute ECE using the L2 norm whereas *[Mukhoti et al., 2020]* uses the L1 norm.  Our choice of measuring L2 ECE is in line with more recent work in calibration error (*[Kumar et al., 2018, Roelofs et al., 2020]*). In particular, similar to *[Roelofs et al., 2020]*, we use L2 error rather than L1 since L2 increases the sensitivity of the error metric to extremely poorly calibrated predictions, which tend to be more harmful in applications. The lower # epochs do not affect the results. We do not see improvement in accuracy or ECE after epoch #150.
> 6. .
>     1. **What about multiclass ECE?** Reducing top-label calibration error using calibration-incentivizing training loss functions is the focus of our work, similar to *[Kumar et al., 2018]*, *[Krishnan and Tickoo, 2020]* and *[Mukhoti et al., 2020]*. Whereas marginal ECE *[Kumar et al., 2019]* is also important in many applications, top-label ECE is still the most popular one being studied. Nevertheless, we have now also evaluated marginal ECE and observed that our methods (which are trained for soft top-label ECE) yield the best marginal ECE numbers as well (see table below).
>   The terms in S-AvUC [equation 16] are tied to the top-label and this does readily lend itself to a multiclass extension. The soft binning approach however immediately yields a trainable soft version of marginal ECE analogous to the top-label case. We have now noted this in the paper.
>   |   (CIFAR-100)  | Marginal ECE | Top-Label ECE (from paper) |
> |:--------------:|:------------:|:--------------------------:|
> |       NLL      |     3.78%    |            9.10%           |
> |   NLL + MMCE   |     3.61%    |            4.77%           |
> |      Focal     |     4.35%    |            4.66%           |
> | Focal + SB-ECE |     3.05%    |            2.30%           |
> | Focal + S-AvUC |     2.72%    |            1.57%           |
>     2. **Comparison to post-hoc calibration methods** Post-hoc calibration techniques can be used complementary to ours and for this reason, we do not compare our techniques to these, similar to *[Krishnan and Tickoo, 2020, Kumar et al., 2018]*. Nevertheless, after your and reviewer 2pg6's suggestions we did run a comparison of our techniques against Dirichlet calibration *[Kull et al, 2019]* and noted that our techniques outperformed it. The results can be seen in bullet point 3 of our response to reviewer 2pg6.
> 7.  **Why is the uncalibrated model sometimes better than TS-SB-ECE in Figure S1?** The core goal of Figure S1 is to demonstrate that TS-SB-ECE is an improvement over TS. All the 6 plots show this and this ends up being the case for 35 of the 42 cases in Table S1.
>   It is a valid and previously reported *[Ovadia et al., 2019]* observation however that TS itself can sometimes make ECE worse, especially under distribution shift. When TS improves ECE, TS-SB-ECE outperforms it. When TS makes ECE worse, TS-SB-ECE still reduces ECE in a few cases, such as for CIFAR-100 with MSE in Figure S1.
>   Finally, we note that on the test set, in most cases, TS-SB-ECE outperforms the uncalibrated model (31 of 42 cases, Table S1). Even under distribution shift, TS-SB-ECE outperforms the uncalibrated model in most cases.
>
> References:
>   See response to reviewer 2pg6

---

> > ### Comment · Reviewer_CL78 · 2021-08-25
> > **Response to Authors**
> >
> > I would like to thank the authors for their response and addressing most of my concerns. Please include the way the values in eq. 8-10 are computed in the main paper.  Is batch-size an important factor in the calibration accuracy of the methods? Thanks for clarifying the pros and cons of TS-SB-ECE and TS-HB-ECE. Regarding the costs comparison, although it is not of primary importance, the TS-SB-ECE requires to compute gradients and maybe for multiple epochs; it might still introduce some overhead.

---

> > > ### Author Response · Authors · 2021-08-26
> > > **Thanks for the review and for the follow-up responses**
> > >
> > > Thanks for your review and for your responses to our rebuttal. This helped us improve the paper greatly. We will make sure to address all your suggestions in the paper.
> > >
> > > You make a good point about the cost comparison for TS-SB-ECE and TS-HB-ECE. There can be multiple epochs and computing gradients has some overhead, although it is worth noting that gradients need not be calculated for the full network but only for the top layer (specifically w.r.t. parameter T).
> > >
> > > You bring up another excellent question regarding batch size. In our experiments we used a per-core-batch-size of 64 on a 2x2 TPU topology with 8 cores for an effective batch size of 512. Both the baseline runs and experimental runs (all runs from Table S1) used this batch size. In general, larger batch size is better for the computation of soft calibration losses, but we did not see significant ECE gains with a further increase in batch size. We have now made a note of this in the 'hyperparameter tuning' section in the appendix.

---

### Official Review · Reviewer_2pg6 · 2021-07-20

**Rating:** 6
**Confidence:** 5

**Summary:**

The paper proposes a soft-binning scheme to compute calibration metric so that the models can be trained using SGD to minimize the calibration error together with the classification error. Two metrics have been tried (ECE and AvUC) and the results show improved performance on image classification datasets.

**Limitations And Societal Impact:**

Adequate

**Main Review:**

## Post-rebuttal update
The authors have clarified both of my concerns about the paper. As mentioned in the discussion below, I would strongly encourage the authors to include the discussion on train ECE overfitting and the role of interleaved training. I'm increasing the score.

## Strengths
- The idea of soft-binning to make the calibration metric trainable is interesting and may have some value.
- Overall the paper is well written.

## Weaknesses
- The main issue of miscalibration is due to overfitting in the training set (wrt to calibration) [Guo et al]. So it is not clear to me that minimize calibration error on the training set would solve the miscalibration issue on the test set. Note that it has been observed that the modern networks have perfect calibration on the training set (if a proper loss such as CE is minimized) but miscalibrated on an unseen test set [Mukhoti et al]. This is a major concern and the marginal improvements in performance may be due to some artifacts of the training scheme. Please clarify this.

- Since the method is optimized for ECE it would make sense to measure calibration error using a different metric such as KS-error [Gupta et al]. This is important to show that the method is not overfitted to ECE in some way. Also, KS-error does not have any issues related to binning.

- In Table 1 latest methods need to be compared such as [Kull et al, 2019, Gupta et al].

**Time Spent Reviewing:**

4

---

> ### Author Response · Authors · 2021-08-11
> **Response to Reviewer 2pg6**
>
> Thanks for so many great suggestions. We have addressed all of them and made multiple additions to our paper.
>
> 1. .
>     1. **Why incentivize lower train ECE which NLL already overfits?** The concerns raised here are indeed valid - the primary loss even by itself (especially NLL) can heavily overfit to the train ECE *[Mukhoti et al., 2020]*. This is an important consideration, and one which we paid close attention to during our experiments. We saw this happen on 2 of our 7 dataset + primary loss settings (CIFAR-100/10 datasets with the NLL primary loss; see table below with L1 norm ECE). For these two cases only, our hypothesis was that this could be fixed by doing “interleaved training”. Each epoch is split into two - the first part optimizes NLL on the majority train set and the second part optimizes soft calibration objectives on the “held out” train set. This way we avoid incentivizing something that is already overfit. The second part of each epoch incentivizes the predicted distribution to have higher entropy. The hypothesis was validated: soft calibration objectives with interleaved training helped reduce test ECE relative to baseline on CIFAR-100 + NLL as can be seen in the numbers from table S1 in the appendix. In this way, soft calibration objectives can help improve calibration even when the primary loss overfits train ECE. We have now clarified this in the appendix.
>   We also note that in practice in many cases, ECE does not heavily overfit the train set and reducing train ECE does help to reduce test ECE.  This was the case in 5 of our 7 dataset + primary loss settings (see table below with L1 norm ECE). The *[Krishnan and Tickoo, 2020]* and the *[Kumar et al., 2018]* papers also incentivize the model to reduce ECE on the train set using secondary losses and show significant benefits of this for reducing test and OOD ECE. Our experiments also show the same.
>   | (Dataset + Loss) | Test ECE | Train ECE | Test ECE / Train ECE |
> |:-------------------------------:|:--------:|:---------:|:--------------------:|
> |         CIFAR-100 + NLL         |   6.88%  |   0.45%   |         15.14        |
> |        CIFAR-100 + Focal        |   4.21%  |   9.18%   |         0.46         |
> |         CIFAR-100 + MSE         |   3.51%  |   2.49%   |         1.41         |
> |          CIFAR-10 + NLL         |   2.66%  |   0.05%   |         49.82        |
> |         CIFAR-10 + Focal        |   5.28%  |   7.13%   |         0.74         |
> |          CIFAR-10 + MSE         |   6.53%  |   9.82%   |         0.66         |
> |          Imagenet + NLL         |   3.35%  |   4.34%   |         0.77         |
>
>     2. **Is this marginal improvement?** We have now run evaluations to measure the effect size (Cohen’s d-value, *[Wiki. Cohen D]*) of each secondary loss for reducing ECE across datasets and primary losses (see table below). We observe that among all secondary losses, S-AvUC results in the most effective ECE reduction both without temperature scaling (d=7.47) and with our recommended temperature scaling variant (d=4.86). Note that d>2.0 is considered a huge effect *[Wiki. Cohen D]*.
>   | (Cohen’s d-value) | Without TS-SB-ECE | With TS-SB-ECE |
> |:-----------------:|:-----------------:|:--------------:|
> |        MMCE       |        6.26       |      3.19      |
> |        AvUC       |        3.87       |      1.27      |
> |       SB-ECE      |        5.10       |      2.43      |
> |       S-AvUC      |        7.47       |      4.86      |
>
> 2. **KS-error as calibration measure:** Thanks for this insight. We have now evaluated all our models in terms of KS-error *[Gupta et al., 2020]* and the mean-sweep calibration error (ECE-sweep) *[Roelofs et al., 2020]*. Table 1a augmented with these error measures is below and shows that soft calibration objectives far outperform existing methods on CIFAR-100. These measurements further validate our claims. It is also worth noting that even before these additions, we were training for either equal-width soft ECE or soft AvUC and evaluating on equal-mass ECE. All these observations definitively confirm that this is not a case of overfitting to an objective. These measures have been added to the appendix.
>   |   (CIFAR-100)  | KS-error | ECE-sweep | ECE (from paper) |   |
> |:--------------:|:--------:|:---------:|:----------------:|---|
> |       NLL      |   6.71%  |   9.02%   |       9.10%      |   |
> |   NLL + MMCE   |   1.45%  |   4.07%   |       4.77%      |   |
> |      Focal     |   4.19%  |   4.57%   |       4.66%      |   |
> | Focal + SB-ECE |   0.71%  |   2.21%   |       2.30%      |   |
> | Focal + S-AvUC |   0.53%  |   1.24%   |       1.57%      |   |
>
> 3. .
>     1. **Comparison to Dirichlet calibration:** Thanks for the suggestion. In Table 1, we have compared our methods to existing calibration-incentivizing losses (Focal, MMCE) that operate during training. AvUC loss also falls in this category but is skipped in Table 1 (features in table S1) because in general it does not seem to help in the non-SVI setting.
> The *[Kull et al. 2019, Gupta et al., 2020]* techniques are post-hoc calibration methods. There is a whole suite of these techniques which can be used post-training regardless of which loss is used during training. Those techniques can be used complementary to ours and for this reason, we do not compare our techniques to these. Note that the same approach is taken by *[Krishnan and Tickoo, 2020]* which compares Focal loss only to MMCE and Label Smoothing (w/ and w/o temperature scaling) and by *[Kumar et al., 2018]* which compares MMCE loss only to baseline (w/ and w/o temperature scaling).
>   Nevertheless, after your comment we did run a comparison of our techniques against Dirichlet calibration *[Kull et al, 2019]* and noted that our techniques outperformed the results with Dirichlet calibration. Table 1b with a comparison to the best-performing Dirichlet added is as follows:
>   |    (CIFAR-10)   |  ECE  |   |   |   |
> |:---------------:|:-----:|:-:|:-:|---|
> |       NLL       | 5.59% |   |   |   |
> |    NLL + MMCE   | 1.55% |   |   |   |
> | NLL + Dirichlet | 2.45% |   |   |   |
> |      Focal      | 5.45% |   |   |   |
> |  Focal + SB-ECE | 1.19% |   |   |   |
> |  Focal + S-AvUC | 1.58% |   |   |   |
>     2. **Comparison to Dirichlet calibration (Continued):** Dirichlet when combined with each primary loss works as follows:
>   |     (CIFAR-10)    |  ECE  |
> |:-----------------:|:-----:|
> |  NLL + Dirichlet  | 2.45% |
> | Focal + Dirichlet | 2.48% |
> |  MSE + Dirichlet  | 4.69% |
>
> References for all responses:
> * *[Krishnan and Tickoo, 2020]* R. Krishnan and O. Tickoo. Improving model calibration with accuracy versus uncertainty optimization. ArXiv, abs/2012.07923, 2020.
> * *[Kumar et al., 2018]* A. Kumar, Sunita Sarawagi, and Ujjwal Jain. Trainable calibration measures for neural networks from kernel mean embeddings. In ICML, 2018.
> * *[Mukhoti et al., 2020]* Jishnu Mukhoti, Viveka Kulharia, Amartya Sanyal, S. Golodetz, P. Torr, and P. Dokania. Calibrating deep neural networks using focal loss. ArXiv, abs/2002.09437, 2020.
> * *[Roelofs et al., 2020]* R. Roelofs, N. Cain, Jonathon Shlens, and M. Mozer. Mitigating bias in calibration error estimation. ArXiv, abs/2012.08668, 2020.
> * *[Gupta et al., 2020]* Gupta, Kartik, et al. "Calibration of neural networks using splines." arXiv preprint arXiv:2006.12800 (2020).
> * *[Kull et al. 2019]* Kull, Meelis, et al. "Beyond temperature scaling: Obtaining well-calibrated multiclass probabilities with Dirichlet calibration." arXiv preprint arXiv:1910.12656 (2019).
> * *[Guo et al., 2017]* Chuan Guo, Geoff Pleiss, Yu Sun, and Kilian Q. Weinberger. On calibration of modern neural networks. ArXiv, abs/1706.04599, 2017.
> * *[Kumar et al., 2019]* Ananya Kumar, Percy Liang, and Tengyu Ma. Verified uncertainty calibration. In NeurIPS, 2019.
> * *[Ovadia et al., 2019]* Yaniv Ovadia, E. Fertig, J. Ren, Zachary Nado, D. Sculley, S. Nowozin, Joshua V. Dillon, Balaji Lakshminarayanan, and Jasper Snoek. Can you trust your model’s uncertainty? Evaluating predictive uncertainty under dataset shift. In NeurIPS, 2019
> * *[Wiki. Cohen D]* https://en.wikipedia.org/wiki/Effect_size#Cohen's_d
> * *[AWS TS Presc.]* https://docs.aws.amazon.com/prescriptive-guidance/latest/ml-quantifying-uncertainty/temp-scaling.html

---

> > ### Comment · Reviewer_2pg6 · 2021-08-23
> > **"Interleaved training" not described in the original submission**
> >
> > I thank the authors for their detailed response which addressed most of my concerns.
> >
> > However, the main concern of overfitting on the training set seems to have been tackled using "interleaved training" which has not been mentioned in the original submission. To me, this is a significant component of the method and it is not clear if it is added in the rebuttal phase. Please clarify this.
> >
> > This raises some concerns on which component yield the most improvement,  interleaved training or the soft-calibration objectives. In my opinion, the narrative of the paper needs to be substantially modified to clarify this.

---

> > > ### Author Response · Authors · 2021-08-23
> > > **Interleaving was not performed during rebuttal and is not needed for any of our best results**
> > >
> > > Thanks for your response.
> > >
> > > **Is interleaving a significant component of the method?** To emphasize, we do not actually observe overfitting to train ECE in most of our experiments (5 of 7 dataset + primary-loss settings). *These 5 settings (with no interleaving in play) account for our best results for all 3 datasets: CIFAR-10, CIFAR-100 and Imagenet.* We did see overfitting to train ECE for CIFAR-10/100 + NLL. Note however that the NLL primary loss baseline for CIFAR-10/100 is itself significantly worse than the Focal primary loss baseline [Mukhoti et al., 2020]. Instead of following papers such as [Mukhoti et al., 2020] and leaving the CIFAR NLL baseline as weak, we tried to improve these results. We observed that primary loss was overfitting to train ECE in this case and we tried (successfully) interleaving as an aside to improve ECE in this setting.
> > >
> > > **Was interleaving added during rebuttal?** No. Interleaving is a strategy we applied during our original submission to improve the weak results for CIFAR-10/100 + NLL. The implementation we did during rebuttal was 1) to add new metrics (KS-Error, Marginal ECE, ECE-Sweep) as suggested to validate our results and 2) to compare against a new baseline (Dirichlet calibration). Because we found no evidence of overfitting on the most competitive primary losses, we inadvertently omitted interleaving. Overfitting to train ECE is a scenario many researchers would immediately consider, so we have now directly referenced it in the main article with a new section in appendix.
> > >
> > > **Which component (interleaving or soft calibration objectives) yields the most improvement?** Interleaved training without the addition of soft calibration training objectives yields no ECE improvement as one would expect. If both parts of each epoch train for the same objective, this is no different from the standard training procedure. The main observation which all our results consistently validate is that training using soft calibration objectives on a dataset reduces ECE on that dataset. All the actual ECE reduction then comes from the soft calibration objectives and “whether to interleave” is a boolean hyperparameter, a detail of our method, which ends up being useful when ECE is already overfit to the train dataset. Even then, it is not useful to obtain our best results. As stated above, this has now been clarified in the appendix.

---

> > > > ### Comment · Reviewer_2pg6 · 2021-08-25
> > > > **Thanks for the clarification**
> > > >
> > > > Thanks for the clarification. So it seems that if the train ECE is not overfitted (for example when the focal loss is used) then the introduced soft-calibration objectives are useful and improve calibration even without interleaved training. In cases where train ECE overfits, one would require a mechanism such as interleaved training along with soft-calibration objectives. I would strongly encourage include this discussion in the paper.

---

> > > > > ### Author Response · Authors · 2021-08-26
> > > > > **Thanks for the review and for the follow-up responses**
> > > > >
> > > > > Thanks for your review and for your responses to our rebuttal. This helped us improve the paper greatly. We will make sure to add the discussion you suggested in the paper.

---

### Decision · Program_Chairs · 2021-09-27

**Decision:**

Accept (Poster)

**Comment:**

The authors' response has addressed all the reviewers' concerns, and all the reviewers' final score was "weak accept", which is borderline. It seems that main issue was that they were not completely sure how the different design choices are related to the final performance, which would make this method somewhat less useful to practitioners. For example, Reviewer CL78 summarized:

It seems incorporating the soft-calibration objectives helps to improve the calibration during training. I am not getting a higher score because most of the paper is devoted to SB-ECE, while the most successful secondary loss is S-AvUC. However, there is not much theoretical support (except the previous work under the SVI setting) for different design choices made by the authors (I am not much familiar with SVI). Furthermore, there are some combinations of primary losses that have a great impact on the final results. Although the soft-calibration objectives can improve the calibration in most cases, it might not be easy for a practitioner to summarize which combinations of primary and secondary loss should be selected in advance.